# Graph Diffusion Transformers for Multi-Conditional Molecular Generation

**Gang Liu, Jiaxin Xu, Tengfei Luo, Meng Jiang**
University of Notre Dame
{gliu7, jxu24, tluo, mjiang2}@nd.edu

## Abstract

Inverse molecular design with diffusion models holds great potential for advancements in material and drug discovery. Despite success in unconditional molecular generation, integrating multiple properties such as synthetic score and gas permeability as condition constraints into diffusion models remains unexplored. We present the Graph Diffusion Transformer (Graph DiT) for multi-conditional molecular generation. Graph DiT integrates an encoder to learn numerical and categorical property representations with the Transformer-based denoiser. Unlike previous graph diffusion models that add noise separately on the atoms and bonds in the forward diffusion process, Graph DiT is trained with a novel graph-dependent noise model for accurate estimation of graph-related noise in molecules. We extensively validate Graph DiT for multi-conditional polymer and small molecule generation. Results demonstrate the superiority of Graph DiT across nine metrics from distribution learning to condition control for molecular properties. A polymer inverse design task for gas separation with feedback from domain experts further demonstrates its practical utility.

## 1 Introduction

Diffusion models for molecular graphs are essential for inverse design of materials and drugs by generating molecules and polymers (macro-molecules) [40, 46], because the models can be effectively trained to predict discrete graph structures and atom/bond types in denoising processes [43]. Practical inverse designs consider multiple factors such as molecular synthetic score and various properties [15], known as the task of multi-conditional graph generation.

Existing work converted multiple conditions into a single one and solved the task as single-condition generation [5, 25]. However, multi-property relations may not be properly or explicitly defined [5]. First, the properties have diverse scales and units. For example, the synthetic complexity ranges from 1 to 5 [8], while the gas permeability varies widely, exceeding 10,000 in Barrier units [4]. This gap makes it hard for models to balance the conditions. Second, multi-conditions consist of a mix of categorical and numerical properties. The common practice of addition [47] or multiplication [25] is inadequate for combination.

Figure 1(a) empirically illustrates the challenges in multi-conditional generation, i.e., discovering molecules meeting multiple properties. We used a test set of 100 data points with three properties: synthesizability (Synth.) [12], $O_2$ and $N_2$ permeability ($O_2$Perm and $N_2$Perm) [4]. A single-conditional diffusion model generated up to 30 graphs for each condition, resulting in a total of 90 graphs for three conditions. We sort the 30 graphs in each set using a polymer property Oracle (see appendix B.3). Then, we check whether a shared polymer structure that meets multi-property constraints can be identified across different condition sets. If we find the polymer, its rank $K$ (where $K$ is between 1 and 30) indicates how high it appears on the lists, considering all condition sets. If not, we set $K$ as 30. Figure 1(a) shows the frequency distribution of $K$ on the 100 test cases. The median $K$ was

38th Conference on Neural Information Processing Systems (NeurIPS 2024).

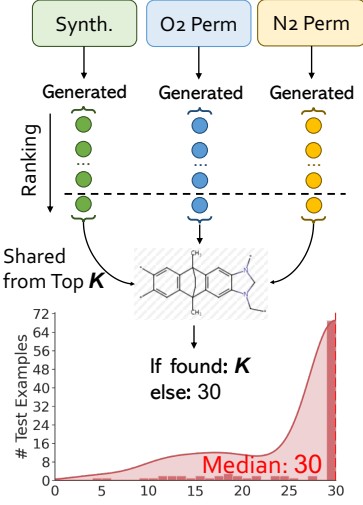

(a) **Existing work's limitation**: A median rank of 30 showed that on fewer than half test polymers, the sets of generated graphs from different single conditions intersected, indicating a failure to generate polymers meeting multiple properties.

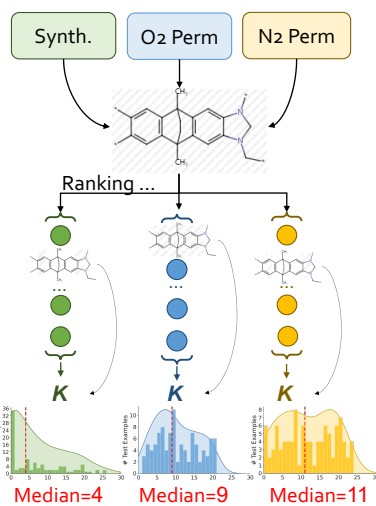

(b) **Proposed work**: Our idea, multi-conditional guidance for diffusion models, successfully generated polymers that satisfied multi-property constraints. It achieved a higher rank than 30 in any set of the single-conditional generated graphs.

Figure 1: Multi-conditional diffusion guidance in (b) generates polymers of higher property accuracy than existing work in (a). Explanations are in Section 1 and details are in appendix B.3.

30, indicating that the multiple properties were not met on over half of the test polymers despite generating a large number of graphs.

To address these challenges, we project multi-properties into representations by *learning*, thereby guiding the diffusion process for molecule generation. We propose the Graph Diffusion Transformer (Graph DiT) for graph denoising under conditions. Graph DiT has a condition encoder for property representation learning and a graph denoiser. The condition encoder utilizes a novel clustering-based method for numerical properties and one-hot encoding for categorical ones to learn multi-property representations. The graph denoiser first integrates node and edge features into graph tokens, then denoise these tokens with adaptive layer normalization (AdaLN) in Transformer layers [19, 34]. AdaLN replaces the molecular statistics (mean and variance) in each hidden layer with those from the condition representation, effectively outperforming other predictor-based and predictor-free conditioning methods [22, 43, 34], as shown in Section 4.4. We observe that existing forward diffusion processes [43, 22] apply noise separately to atoms and bonds, which may compromise the accuracy of Graph DiT in noise estimation. Hence, we propose a novel graph-dependent noise model that effectively applies noise tailored to the dependencies between atoms and bonds within the graph.

Results in Figure 1(b) show that the polymers generated by Graph DiT closely align with multi-property constraints. For each test case, we have *one* graph generated from Graph DiT conditional on three properties. The Oracle determines the rank of this graph among 30 single-conditionally generated graphs for each condition. We find the median ranks are 4, 9, and 11, for Synth., $O_2$ Perm, and $N_2$ Perm, respectively, all much higher than 30. Note that the ranked set of 30 graphs was very competitive because the model was trained on the specific condition dedicatedly.

In experiments, we evaluate model performance on one polymer and three small molecule datasets. The polymer dataset includes four numerical conditions for multi-conditional evaluation. Our model has the lowest average mean absolute error (MAE), significantly reducing the error by 17.86% compared to the best baseline. It also excels in small molecule tasks, achieving over 0.9 accuracy on task-related categorical conditions, notably surpassing the baseline accuracy of less than 0.6. We also examine the model's utility in inverse polymer designs for $O_2/N_2$ gas separation, with domain expert feedback highlighting our model's practical utility in multi-conditional molecular design.

## 2 Problem Definition

### 2.1 Multi-Conditional Inverse Molecular Design

A molecular graph $G = (V, E)$ consists of a set of nodes (atoms) $V$ and edges (bonds) $E$. We follow [43] and define "non-bond" as a type of edge. There are $N$ atoms and each atom has a one-hot encoding, denoting the atom type. We represent it as $\mathbf{X}_V \in \mathbb{R}^{N \times F_V}$, where $F_V$ is the total number of atom types. Similarly, the bond features are a tensor $\mathbf{X}_E \in \mathbb{R}^{N \times N \times F_E}$, representing both the graph structure and $F_E$ bond types.

Let $\mathcal{C} = \{c_1, c_2, \ldots, c_M\}$ be a set of $M$ numerical and categorical conditions. The task is: $q(G \mid c_1, c_2, \ldots, c_M) \propto q(G)q(c_1, c_2, \ldots, c_M \mid G)$, where $q$ represents observed probability. We use a model parameterized by $\theta$ for multi-conditional molecular generation $p_\theta(G \mid \mathcal{C})$. The evaluation involves both distribution learning $q(G)$ [35] and condition control $q(c_1, c_2, \ldots, c_M \mid G)$. We follow previous work in assuming that there exist different oracle functions $\mathcal{O}$ that can independently evaluate each conditioned property [14]: $q(c_1, c_2, \ldots, c_M \mid G) = \prod_{i=1}^{M} \mathcal{O}_i(c_i \mid G)$. Note that the oracles are *not* used in the training of $p_\theta$.

### 2.2 Diffusion Model on Graph Data

Diffusion models consist of forward and reverse diffusion processes [17]. We refer to the forward diffusion process as the diffusion process following [17]. The diffusion process $q(G^{1:T} \mid G^0) = \prod_{t=1}^{T} q(G^t \mid G^{t-1})$ corrupts molecular graph data ($G^0 = G$) into noisy states $G^t$. As timesteps $T \to \infty$, $q(G^T)$ converges a stationary distribution $\pi(G)$. The reverse Markov process $p_\theta(G^{0:T}) = q(G^T) \prod_{t=1}^{T} p_\theta(G^{t-1} \mid G^t)$, parameterized by neural networks, gradually denoises the latent states toward the desired data distribution.

**Diffusion Process**  One may perturb $G^t$ in a discrete state-space to capture the structural properties of molecules [43]. Two transition matrices $\mathbf{Q}_V \in \mathbb{R}^{F_V \times F_V}$ and $\mathbf{Q}_E \in \mathbb{R}^{F_E \times F_E}$ are defined for nodes $\mathbf{X}_V$ and edges $\mathbf{X}_E$, respectively [43]. Then, each step $q(G^t \mid G^{t-1}, G^0) = q(G^t \mid G^{t-1})$ in the diffusion process is sampled as follows.

$$\begin{cases} q(\mathbf{X}_V^t \mid \mathbf{X}_V^{t-1}) = \mathrm{Cat}\left(\mathbf{X}_V^t; \mathbf{p} = \mathbf{X}_V^{t-1}\mathbf{Q}_V^t\right), \\ q(\mathbf{X}_E^t \mid \mathbf{X}_E^{t-1}) = \mathrm{Cat}\left(\mathbf{X}_E^t; \mathbf{p} = \mathbf{X}_E^{t-1}\mathbf{Q}_E^t\right), \end{cases} \tag{1}$$

where $\mathrm{Cat}(\mathbf{X}; \mathbf{p})$ denotes sampling from a categorical distribution with probability $\mathbf{p}$. We remove the subscript $_{(V/E)}$ when the description applies to both nodes and edges. It is assumed that the noise $\mathbf{Q}^i$ ($i \leq t$) is independently applied to $\mathbf{X}$ in each step $i$, allowing us to rewrite $q(\mathbf{X}^t \mid \mathbf{X}^{t-1})$ as the probability of the initial state $q(\mathbf{X}^t \mid \mathbf{X}^0) = \mathrm{Cat}\left(\mathbf{X}^t; \mathbf{p} = \mathbf{X}^0\bar{\mathbf{Q}}^t\right)$, where $\bar{\mathbf{Q}}^t = \prod_{i \leq t} \mathbf{Q}^i$.

**Noise Scheduling**  Transition matrices $\mathbf{Q}_V$ and $\mathbf{Q}_E$ control the noise applied to atom features and bond features, respectively. Vignac et al. [43] defined $\pi(G) = (\mathbf{m}_X \in \mathbb{R}^{F_V}, \mathbf{m}_E \in \mathbb{R}^{F_E})$ as the marginal distributions of atom types and bond types. The transition matrix at timestep $t$ is $\mathbf{Q}^t = \alpha^t\mathbf{I} + (1 - \alpha^t)\mathbf{1}\mathbf{m}'$ for atoms or bonds, where $\mathbf{m}'$ denotes the transposed row vector. Therefore, we have $\bar{\mathbf{Q}}^t = \bar{\alpha}^t\mathbf{I} + (1 - \bar{\alpha}^t)\mathbf{1}\mathbf{m}'$, where $\bar{\alpha}^t = \prod_{\tau=1}^{t} \alpha^\tau$. The cosine schedule [32] is often chosen for $\bar{\alpha}^t = \cos(0.5\pi(t/T + s)/(1 + s))^2$.

**Reverse Process**  With the initial sampling $G^T \sim \pi(G)$, the reverse process generates $G^0$ iteratively in reversed steps $t = T, T-1, \ldots, 0$. We use a neural network to predict the probability $p_\theta(\tilde{G}^0 \mid G^t)$ as the product over nodes and edges [1, 43]:

$$p_\theta(\tilde{G}^0 \mid G^t) = \prod_{v \in V} p_\theta(v^{t-1} \mid G^t) \prod_{e \in E} p_\theta(e^{t-1} \mid G^t) \tag{2}$$

$p_\theta(\tilde{G}^0 \mid G^t)$ could be combined with $q(G^{t-1} \mid G^t, G^0)$ to estimate the reverse distribution on the graph $p_\theta(G^{t-1} \mid G^t)$. For example, $p_\theta(v^{t-1} \mid G^t)$ is marginalized over predictions of node types $\tilde{v} \in \tilde{\mathbf{x}}_v$, which applies similarly to edges:

$$p_\theta(v^{t-1} \mid G^t) = \sum_{\tilde{v} \in \tilde{\mathbf{x}}_v} q(v^{t-1} \mid \tilde{v}, G^t)p_\theta(\tilde{v} \mid G^t). \tag{3}$$

The neural network could be trained to minimize the negative log-likelihood [43].

$$L = \mathbb{E}_{q(G^0)}\mathbb{E}_{q(G^t \mid G^0)}\left[-\mathbb{E}_{\mathbf{x} \in G^0} \log p_\theta\left(\mathbf{x} \mid G^t\right)\right] \tag{4}$$

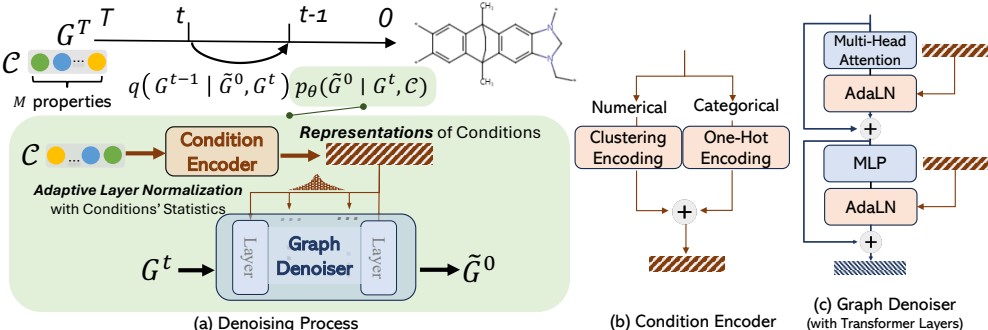

Figure 2: Denoising framework and architectures for Graph DiT. Details are in Section 3.2.

where $\mathbf{x} \in G^0$ denotes the node or edge features. Typically, the reverse process in diffusion models does not consider molecular properties as conditions. While there have been efforts to introduce property-related guidance using additional predictors, the more promising approach of predictor-free guidance [16], particularly in multi-conditional generation, remains underexplored.

# 3 Multi-Conditional Graph Diffusion Transformers

We present the denoising framework of Graph DiT in Figure 2. The condition encoder learns the representation of $M$ conditions. The statistics of this representation like mean and variance are used to replace the ones from the molecular representations [19] (see Section 3.2). Besides, we introduce a new noise model in the diffusion process to better fit graph-structured molecules (see Section 3.1).

## 3.1 Graph-Dependent Noise Models

The transition probability of a node or an edge should rely on the joint distribution of nodes and edges in the prior state. However, as an example shown in Eq. (1), current diffusion models [22, 43, 25] treat node and edge state transitions as independent, misaligning with the denoising process in Eq. (3). This difference between the sampling distributions of noise in the diffusion and reverse processes introduces unnecessary challenges to multi-conditional molecular generations.

To address this, we use a single matrix $\mathbf{X}_G \in \mathbb{R}^{N \times F_G}$ to represent graph tokens for $G$, with $F_G = F_V + N \cdot F_E$. Token representations are created by concatenating the node feature matrix $\mathbf{X}_V$ and the flattened edge connection matrix from $\mathbf{X}_E$. Each row vector in $\mathbf{X}_G$ contains features for both nodes and edges, representing all connections and non-connections. Hence, we could design a transition matrix $\mathbf{Q}_G$ considering the joint distribution of nodes and edges. $\mathbf{Q}_G \in \mathbb{R}^{F_G \times F_G}$ is constructed from four matrices $\mathbf{Q}_V, \mathbf{Q}_{EV} \in \mathbb{R}^{F_E \times F_V}, \mathbf{Q}_E, \mathbf{Q}_{VE} \in \mathbb{R}^{F_V \times F_E}$, denoting the transition probability ("dependent old state" $\rightarrow$ "target new state") node $\rightarrow$ node; edge $\rightarrow$ node; edge $\rightarrow$ edge; node $\rightarrow$ edge, respectively.

$$\mathbf{Q}_G = \begin{bmatrix} \mathbf{Q}_V & \mathbf{1}'_N \otimes \mathbf{Q}_{VE} \\ \mathbf{1}_N \otimes \mathbf{Q}_{EV} & \mathbf{1}_{N \times N} \otimes \mathbf{Q}_E \end{bmatrix}, \tag{5}$$

where $\otimes$ denotes the Kronecker product, $\mathbf{1}_N, \mathbf{1}'_N$, and $\mathbf{1}_{N \times N}$ represent the column vector, row vector, and matrix with all 1 elements, respectively. According to Eq. (5), the first $F_V$ columns in $\mathbf{Q}_G$ determine the node feature transitions based on both node features (first $F_V$ rows) and edge features (remaining $N \cdot F_E$ rows). Conversely, the remaining $N \cdot F_E$ columns determine the edge feature transitions, depending on the entire graph. We introduce a new diffusion noise model:

$$q(\mathbf{X}_G^t \mid \mathbf{X}_G^{t-1}) = \widetilde{\mathrm{Cat}}\left(\mathbf{X}_G^t; \tilde{\mathbf{p}} = \mathbf{X}_G^{t-1}\mathbf{Q}_G^t\right), \tag{6}$$

where $\tilde{\mathbf{p}}$ is the unnormalized probability and $\widetilde{\mathrm{Cat}}$ denotes categorical sampling: The first $F_V$ columns of $\tilde{\mathbf{p}}$ are normalized to sample $\mathbf{X}_V^t$, while the remaining $N \cdot E$ dimensions are reshaped and normalized to sample edges $\mathbf{X}_E^t$. These components are combined to form $\mathbf{X}_G^t$, completing the $\widetilde{\mathrm{Cat}}$ sampling.

**Choice of $\mathbf{Q}_{VE}$ and $\mathbf{Q}_{EV}$** Similar to the definitions of $\mathbf{m}_V$ and $\mathbf{m}_E$ [43], we leverage the prior knowledge within the training data for the formulation of task-specific matrices, $\mathbf{Q}_{EV}$ and $\mathbf{Q}_{VE}$. We calculate co-occurrence frequencies of atom and bond types in training molecular graphs to obtain the marginal atom-bond co-occurrence probability distribution. For each bond type, each row in $\mathbf{m}_{EV}$ represents the probability of co-occurring atom types. $\mathbf{m}_{VE}$ is the transpose of $\mathbf{m}_{EV}$ and has a similar meaning. Subsequently, we define $\mathbf{Q}_{EV} = \bar{\alpha}^t\mathbf{I} + (1-\bar{\alpha}^t)\mathbf{1}\mathbf{m}'_{EV}$ and $\mathbf{Q}_{VE} = \bar{\alpha}^t\mathbf{I} + (1-\bar{\alpha}^t)\mathbf{1}\mathbf{m}'_{VE}$.

## 3.2 Denoising Models with Multi-Property Conditions

We present Graph DiT as the denoising model to generate molecules under multi-conditions $\mathcal{C} = \{c_1, c_2, \ldots, c_M\}$ without extra predictors.

**Predictor-Free Guidance** The predictor-free reverse process $\hat{p}_\theta(G^{t-1} \mid G^t, \mathcal{C})$ aims to generate molecules with a high probability $q(\mathcal{C} \mid G^0)$. This could be achieved by a linear combination of the log probability for unconditional and conditional denoising [16]:

$$\hat{p}_\theta(G^{t-1} \mid G^t, \mathcal{C}) = \log p_\theta(G^{t-1} \mid G^t) + s\left(\log p_\theta(G^{t-1} \mid G^t, \mathcal{C}) - \log p_\theta(G^{t-1} \mid G^t)\right), \quad (7)$$

where $s$ denotes the scale of conditional guidance. Unlike classifier-free guidance [16], which typically predicts noise, we directly estimate $p_\theta(\tilde{G}^0 \mid G^t, \mathcal{C})$. We one one denoising model $f_\theta(G^t, \mathcal{C})$ for both $p_\theta(\tilde{G}^0 \mid G^t)$ and $p_\theta(\tilde{G}^0 \mid G^t, \mathcal{C})$. Here, $f_\theta(G^t, \mathcal{C} = \emptyset)$ computes the unconditional probability by substituting the original conditional embeddings with the null value. During training, we randomly drop the condition with a ratio, i.e., $\mathcal{C} = \emptyset$, to learn the embedding of the null value. $f_\theta(G^t = \mathbf{X}_G^t, \mathcal{C})$ comprises two components: the condition encoder and the graph denoiser. An overview of the architecture is presented in Figure 2.

**Condition Encoder** We treat the timestep $t$ as a special condition and follow [31] to obtain a $D$-dimensional representation $\mathbf{t}$ with sinusoidal encoding. For property-related numerical or categorical condition $c_i \in \mathcal{C}$, we apply distinct encoding operations to get $D$-dimensional representation. For a categorical condition, we use the one-hot encoding. For a numerical variable, we introduce a clustering encoding method. This defines learnable centroids, assigning $c_i$ to clusters, and transforming the soft assignment vector of condition values into the representation. It could be implemented using two Linear layers and a Softmax layer in the middle as: $\text{Linear}(\text{Softmax}(\text{Linear}(c_i)))$. Finally, we could obtain the representation of the condition as $\mathbf{c} = \sum_{i=1}^M \text{encode}(c_i)$, where $\text{encode}$ is the specific encoding method based on the condition type. For numerical conditions, we evaluate our proposed clustering-based approach against alternatives like direct or interval-based encodings [28]. As noted in Section 4.4, the clustering encoding outperforms the other methods.

**Graph Denoiser: Transformer Layers** Given the noisy graph at timestep $t$, the graph tokens are first encoded into the hidden space as $\mathbf{H} = \text{Linear}(\mathbf{X}_G^t)$, where $\mathbf{H} \in \mathbb{R}^{N \times D}$. We then adapt the standard Transformer layers [42] with self-attention and multi-layer perceptrons (MLP), but replace the normalization with the adaptive layer normalization (AdaLN) controlled by the representations of the conditions [19, 34]: $\mathbf{H} = \text{AdaLN}(\mathbf{H}, \mathbf{c})$. For each row $\mathbf{h}$ in $\mathbf{H}$:

$$\text{AdaLN}(\mathbf{h}, \mathbf{c}) = \gamma_\theta(\mathbf{c}) \odot \frac{\mathbf{h} - \mu(\mathbf{h})}{\sigma(\mathbf{h})} + \beta_\theta(\mathbf{c}), \quad (8)$$

where $\mu(\cdot)$ and $\sigma(\cdot)$ are mean and variance values. $\odot$ indicates element-wise product. $\gamma_\theta(\cdot)$ and $\beta_\theta(\cdot)$ are neural network modules in $f_\theta(\cdot)$, each of which consists of two linear layers with SiLU activation [11] in the middle. We have a gated variant $\text{AdaLN}_{gate}$ for residuals:

$$\text{AdaLN}_{gate}(\mathbf{h}, \mathbf{c}) = \alpha_\theta(\mathbf{c}) \odot \text{AdaLN}(\mathbf{h}, \mathbf{c}) \quad (9)$$

We apply the zero initialization for the first layer of $\gamma_\theta(\cdot)$, $\beta_\theta(\cdot)$, and $\alpha_\theta(\cdot)$ [34]. There are other options to learn the structure representation from the condition [34]: In-Context conditioning adds condition representation to the structure representation at the beginning of the structure encoder, and Cross-Attention calculates cross-attention between the condition and structure representation. We observe in Section 4.4 that AdaLN performs best among them.

**Graph Denoiser: Final MLP** We have the hidden states $\mathbf{H}$ after the final Transformer layers, the MLP is used to predict node probabilities $\tilde{\mathbf{X}}_V^0$ and edge probabilities $\tilde{\mathbf{X}}_E^0$ at $t = 0$:

$$\tilde{\mathbf{X}}_G^0 = \text{AdaLN}(\text{MLP}(\mathbf{H}), \mathbf{c}). \quad (10)$$

We split the output $\mathbf{X}_G$ into atom and bond features $\tilde{\mathbf{X}}_V^0, \tilde{\mathbf{X}}_E^0$. The first $F_V$ dimensions of $\tilde{\mathbf{X}}_G^0$ represent node type probabilities, and the remaining $N \cdot F_E$ dimensions cover probabilities for $N$ edge types associated with the node, as detailed in Section 3.1.

**Generation to Molecule Conversion** A common way of converting generated graphs to molecules selects only the largest connected component [43], denoted as Graph DiT-LCC in our model. For Graph DiT, we connect all components by randomly selecting atoms. It minimally alters the generated structure to more accurately reflect model performance than Graph DiT-LCC.

Table 1: Multi-Conditional Generation of 10K Polymers: Results on the synthetic score (Synth.) and three numerical properties (gas permeability for $O_2$, $N_2$, $CO_2$). MAE is calculated between the input conditions and the properties of the generated polymers using Oracles. Best results are **highlighted**.

| Model | Validity ↑ (w/o rule checking) | Distribution Learning | | | | Condition Control | | | | |
|---|---|---|---|---|---|---|---|---|---|---|
| | | Coverage ↑ | Diversity ↑ | Similarity ↑ | Distance ↓ | Synth. ↓ | $O_2$Perm ↓ | $N_2$Perm ↓ | $CO_2$Perm ↓ | Avg. MAE ↓ |
| Graph GA | 1.0000 (N.A.) | 11/11 | 0.8828 | 0.9269 | 9.1882 | 1.3307 | 1.9840 | 2.2900 | 1.9489 | 1.8884 |
| MARS | 1.0000 (N.A.) | 11/11 | 0.8375 | 0.9283 | 7.5620 | 1.1658 | 1.5761 | 1.8327 | 1.6074 | 1.5455 |
| LSTM-HC | 0.9910 (N.A.) | 10/11 | 0.8918 | 0.7937 | 18.1562 | 1.4251 | 1.1003 | 1.2365 | 1.0772 | 1.2098 |
| JTVAE-BO | 1.0000 (N.A.) | 10/11 | 0.7366 | 0.7294 | 23.5990 | **1.0714** | 1.0781 | 1.2352 | 1.0978 | 1.1206 |
| DiGress | 0.9913 (0.2362) | 11/11 | 0.9099 | 0.2724 | 22.7237 | 2.9842 | 1.7163 | 2.0630 | 1.6738 | 2.1093 |
| DiGress v2 | 0.9812 (0.3057) | 11/11 | **0.9105** | 0.2771 | 21.7311 | 2.7507 | 1.7130 | 2.0632 | 1.6648 | 2.0479 |
| GDSS | 0.9205 (0.9076) | 9/11 | 0.7510 | 0.0000 | 34.2627 | 1.3701 | 1.0271 | 1.0820 | 1.0683 | 1.1369 |
| MOOD | 0.9866 (0.9205) | 11/11 | 0.8349 | 0.0227 | 39.3981 | 1.4019 | 1.4961 | 1.7603 | 1.4748 | 1.5333 |
| Graph DiT-LCC (Ours) | 0.9753 (0.8437) | 11/11 | 0.8875 | 0.9560 | 7.0949 | 1.3099 | 0.8001 | 0.9562 | 0.8125 | 0.9697 |
| Graph DiT (Ours) | 0.8245 (0.8437) | 11/11 | 0.8712 | **0.9600** | **6.6443** | 1.2973 | **0.7440** | **0.8857** | **0.7550** | **0.9205** |

## 4 Experiment

**RQ1**: We validate the generative power of Graph DiT compared to baselines from molecular optimization and diffusion models in Section 4.2. **RQ2**: We study a polymer inverse design for gas separation in Section 4.3. **RQ3**: We conduct further analysis to examine Graph DiT in Section 4.4.

### 4.1 Experimental Setup

We use datasets with over ten types of atoms and up to fifty nodes in a molecular graph. We include both numerical and categorical properties for drugs and materials, offering a benchmark for evaluation across diverse chemical spaces. Model performance is validated across up to nine metrics, including distribution coverage, diversity, and condition control capacity for various properties.

**Datasets and Input Conditions**   We have one polymer dataset [40] for materials, featuring three **numerical** gas permeability conditions: $O_2$Perm, $CO_2$Perm, and $N_2$Perm. For drug design, we create three class-balanced datasets from MoleculeNet [46]: HIV, BBBP, and BACE, each with a **categorical** property related to HIV virus replication inhibition, blood-brain barrier permeability, or human $\beta$-secretase 1 inhibition, respectively. We have two more **numerical** conditions for synthesizability from synthetic accessibility (SAS) and complexity scores (SCS) [12, 8].

**Evaluation**   We randomly split the dataset into training, validation, and testing (reference) sets in a 6:2:2 ratio. Evaluations are conducted on 10,000 generated examples with metrics [35] (1) molecular validity (Validity); (2) heavy atom type coverage (Coverage); (3) internal diversity among the generated examples (Diversity); (4) fragment-based similarity with the reference set (Similarity); (5) Fréchet ChemNet Distance with the reference set (Distance) [36]; MAE between the generated and conditioned (6) synthetic accessibility score [12] (Synth.); (7)∼(9) MAE/Accuracy for the numerical/categorical task conditions (Property). The evaluation Oracle uses random forest trained on all task-related molecules [14]. Lower MAE or higher accuracy indicates stronger model controllability.

**Baselines**   We select strong and popular molecular optimization baselines from recent studies [14]: Graph-GA [20], MARS [47], JTVAE [21] with Bayesian optimization (JTVAE-BO), LSTM [6] on SMILES with Hill Climbing (LSTM-HC). We include the most recent diffusion models: GDSS[22], DiGress [43], and their conditional version with extra predictors: MOOD [25], and DiGress v2 [43]. We train multi-task predictors using the same architecture for MOOD and DiGress v2 models to provide additional guidance for generation. For molecular optimization, we formulate the condition set of each test data point as a combined goal, minimizing the sum of the normalized errors between generated and input properties. We train a random forest model for each property using the training data to optimize the molecular structure.

### 4.2 RQ1: Multi-Conditional Molecular Generation

We have the observations from Table 1 and Table 2:

**Chemical Validity**   High validity may not accurately represent the model's generative performance if hard-coded rules are introduced in the algorithm. For example, GraphGA could eliminate non-valid molecules during mutation and crossover iterations to achieve perfect validity in the final evaluation. Without rule checking in the generation-to-molecule step, DiGress, GDSS, and MOOD show a marked performance decline, with validity often dropping from 0.99 to below 0.6. In contrast, Graph DiT often maintains over 0.8 validity without any rule-based processing.

Table 2: Multi-Conditional Generation of 10K Small Molecules: Each dataset involves a numerical synthesizability score (Synth.) and a categorical task-specific property. MAE/Accuracy is calculated by comparing input conditions and generated properties. The best number per metric is **highlighted**.

| Tasks | Model | Validity ↑ (w/o rule checking) | Distribution Learning | | | | Condition Control | | Avg. Rank ↓ |
|---|---|---|---|---|---|---|---|---|---|
| | | | Coverage ↑ | Diversity ↑ | Similarity ↑ | Distance ↓ | Synthe. MAE ↓ | Property Acc. ↑ | |
| Synth. & BACE | Graph GA | 1.0000 (N.A.) | 8/8 | 0.8585 | 0.9805 | 7.4104 | 0.9633 | 0.4690 | 6.5000 |
| | MARS | 1.0000 (N.A.) | 8/8 | 0.8338 | **0.8827** | **6.7923** | 1.0123 | 0.5184 | 5.0000 |
| | LSTM-HC | 0.9972 (N.A.) | 8/8 | 0.8146 | 0.7982 | 17.5585 | 0.9207 | 0.5816 | 3.0000 |
| | JTVAE-BO | 1.0000 (N.A.) | 6/8 | 0.6682 | 0.7281 | 30.4696 | 0.9923 | 0.4628 | 7.5000 |
| | DiGress | 0.3511 (0.2858) | 8/8 | 0.8862 | 0.6942 | 24.6560 | 2.0681 | 0.5061 | 8.0000 |
| | DiGress v2 | 0.3546 (0.2680) | 8/8 | 0.8812 | 0.7027 | 25.3270 | 2.3365 | 0.5113 | 7.5000 |
| | GDSS | 0.2879 (0.2589) | 4/8 | 0.8756 | 0.2708 | 46.7539 | 1.6422 | 0.5036 | 7.5000 |
| | MOOD | 0.9947 (0.4502) | 8/8 | **0.8902** | 0.2587 | 44.2394 | 1.8853 | 0.5062 | 7.0000 |
| | Graph DiT-LCC (Ours) | 0.8646 (0.8495) | 8/8 | 0.8240 | 0.8757 | 6.9836 | 0.4053 | 0.9050 | 2.0000 |
| | Graph DiT (Ours) | 0.8674 (0.8495) | 8/8 | 0.8238 | 0.8752 | 7.0456 | **0.3998** | **0.9135** | **1.0000** |
| Synth. & BBBP | Graph GA | 1.0000 (N.A.) | 9/9 | 0.8950 | **0.9509** | **10.1659** | 1.2082 | 0.3015 | 7.5000 |
| | MARS | 1.0000 (N.A.) | 8/9 | 0.8637 | 0.7696 | 10.9791 | 1.2250 | 0.5189 | 6.0000 |
| | LSTM-HC | 0.9990 (N.A.) | 8/9 | 0.8883 | 0.8932 | 16.3904 | 0.9969 | 0.5590 | 4.0000 |
| | JTVAE-BO | 1.0000 (N.A.) | 5/9 | 0.7458 | 0.5821 | 33.5746 | 1.1619 | 0.4958 | 6.0000 |
| | DiGress | 0.6960 (0.4871) | 9/9 | 0.9098 | 0.6805 | 18.6921 | 2.3658 | 0.6536 | 6.5000 |
| | DiGress v2 | 0.6892 (0.4100) | 9/9 | 0.9107 | 0.6336 | 19.4498 | 2.2694 | 0.6531 | 6.5000 |
| | GDSS | 0.6218 (0.5919) | 3/9 | 0.8415 | 0.2672 | 39.9440 | 1.3788 | 0.5037 | 7.0000 |
| | MOOD | 0.8008 (0.5789) | 9/9 | **0.9273** | 0.1715 | 34.2506 | 2.0284 | 0.4903 | 8.5000 |
| | Graph DiT-LCC (Ours) | 0.8657 (0.8505) | 9/9 | 0.8857 | 0.9324 | 11.8587 | 0.3717 | 0.9390 | 2.0000 |
| | Graph DiT (Ours) | 0.8468 (0.8505) | 9/9 | 0.8856 | 0.9329 | 11.8519 | **0.3551** | **0.9417** | **1.0000** |
| Synth. & HIV | Graph GA | 1.0000 (N.A.) | 28/29 | 0.8993 | **0.9661** | **4.4418** | 0.9839 | 0.6035 | 5.0000 |
| | MARS | 1.0000 (N.A.) | 26/29 | 0.8764 | 0.6517 | 7.2893 | 0.9691 | 0.6455 | 4.0000 |
| | LSTM-HC | 0.9994 (N.A.) | 13/29 | 0.9091 | 0.9145 | 7.4659 | 0.9962 | 0.6736 | 3.0000 |
| | JTVAE-BO | 1.0000 (N.A.) | 3/29 | 0.8055 | 0.4173 | 41.9771 | 1.2359 | 0.4850 | 7.5000 |
| | DiGress | 0.4377 (0.3643) | 22/29 | 0.9194 | 0.8562 | 13.0409 | 1.9216 | 0.5335 | 7.5000 |
| | DiGress v2 | 0.5050 (0.4242) | 24/29 | 0.9193 | 0.8476 | 13.3997 | 1.5934 | 0.5331 | 7.5000 |
| | GDSS | 0.6926 (0.6757) | 4/29 | 0.7817 | 0.1032 | 45.3416 | 1.2515 | 0.4830 | 8.5000 |
| | MOOD | 0.2875 (0.2173) | 29/29 | **0.9280** | 0.1361 | 32.3523 | 2.3144 | 0.5106 | 9.0000 |
| | Graph DiT-LCC (Ours) | 0.7635 (0.7415) | 28/29 | 0.8966 | 0.9535 | 5.8790 | **0.3084** | 0.9766 | **1.5000** |
| | Graph DiT (Ours) | 0.7660 (0.7415) | 28/29 | 0.8974 | 0.9575 | 6.0216 | 0.3086 | **0.9777** | **1.5000** |

**Distribution Learning** GraphGA is a simple yet effective baseline for generating in-distribution molecules, e.g., on BBBP and HIV generation datasets. Diffusion model baselines such as DiGress and MOOD could produce diverse molecules but often fail to capture the original data distribution in multi-conditional tasks. Graph DiT shows the competitive performance of diffusion models in fitting complex molecular data distributions. Using fragment-based similarity and neural network-based distance metrics [36], we achieve the best in the polymer task and rank second in the HIV small molecule task, involving up to 11 and 29 types of heavy atoms, respectively.

**Condition Controllability** LSTM-HC surpasses many baselines, achieving lower average MAE on polymer properties and higher rankings on small molecular properties. However, its control over synthetic scores in polymer tasks is relatively poor. Conversely, MARS effectively manages synthetic scores for polymers but exhibits a larger MAE in gas permeability conditions compared to other baselines. GDSS performs well in gas permeability control but underperforms Graph GA and MARS in terms of the synthetic score condition. DiGress v2 and MOOD, although equipped with the predictor guidance, still exhibit limited condition control compared to their unconditional counterparts over polymer and small molecule tasks. These baselines struggle to balance and control multiple conditions in generation. In contrast, Graph DiT significantly improves diffusion models and achieves the best multi-conditional performance in all tasks. In polymer tasks, Graph DiT reduces MAE on all gas permeability conditions, averaging +17.8% improvement over the best baseline LSTM-HC. For small molecule tasks, Graph DiT consistently ranks top-1 in condition controllability with over 0.9 accuracy in categorical conditions. Compared to Graph DiT-LCC, we observe that Graph DiT, which connects all generated graph components, shows better controllability performance due to minimal rule-based post-generation processing.

### 4.3 RQ2: Polymer Inverse Design for Gas Separation

We aim to design polymers with high $O_2$ and low $N_2$ permeability, demonstrating the models' precise control over related properties. Following Robeson [38]'s definition of high-performance polymers based on the $O_2/N_2$ permeability ratio, we selected 16 polymers meeting this criterion from 609 examples as our test/reference set. The remaining data is used for training and validation. Subsequently, we generated 1,000 polymers conditioned on test set labels.

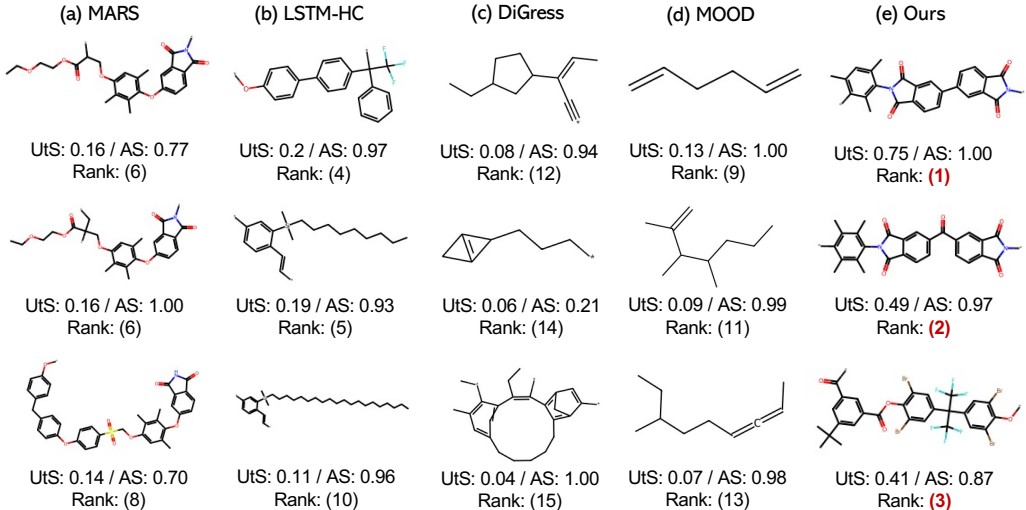

Figure 3: Polymer Inverse Design for $O_2/N_2$ Gas Separation: Feedback from four domain experts includes an average Utility Score (**UtS**) for relative usefulness and an Agreement Score (**AS**) for generated polymers, both ranging [0, 1]. Polymers are generated conditional on {SAS=3.8, SCS=4.3, $O_2$Perm=34.0, $N_2$Perm=5.2}. The top-3 polymers, highlighted, are all generated by Graph DiT.

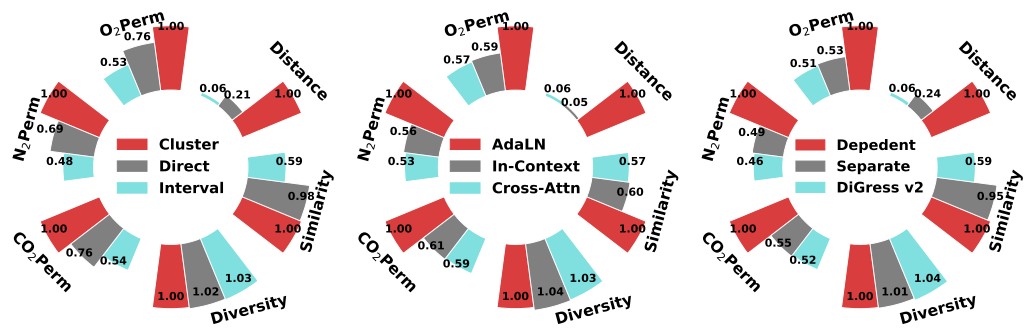

(a) Numerical Condition Encodings  (b) Condition Architectures  (c) Graph Dependent Noise Model

Figure 4: Relative Performance of Different Model Designs: A higher bar indicates better performance. We use the performance of clustering-based encoding or AdaLN as the Reference Value and the current option as the Current Value. Relative performance is calculated as $\frac{\text{Current Value}}{\text{Reference Value}}$ for Similarity and Diversity metrics, and as $\frac{\text{Reference Value}}{\text{Current Value}}$ for other metrics.

In Figure 3, we present the top three polymers generated by each model for a case study with expertise. Initially, a random forest algorithm identifies the top five polymers per method based on average MAE in two gas permeability. These 25 polymers are then shuffled and evaluated by four polymer scientists, who rank them from 1 to 25 using their domain knowledge. Rankings are normalized to a **Utility Score (UtS)** ranging from 0 to 1, with higher scores indicating greater utility. The variance in UtS is converted into an **Agreement Score (AS)** for further evaluation. As shown in Figure 3, there is a high consensus among experts that the three polymers generated by Graph DiT are the most promising for successful polymer inverse design tasks. More details are in appendix D. By comparing generated examples from different models, we have further observations:

- DiGress and MOOD struggle to capture polymerization points, marked with asterisks ("*"), which is one of the most important features that distinguish polymers from small molecules. Additionally, the two methods frequently feature excessive carbon atoms and overly large cycles. These molecular configurations with significant distortion from the canonical geometry of stable compounds may lead to poor synthesizability [35, 7].
- LSTM-HC may result in too-long carbon chains with limited diversity. MARS produces examples with asymmetrical graph structures, challenging polymer synthesis [10, 2].

- Graph DiT generates structurally diverse and symmetric polymers with two polymerization points, indicative of more valid and synthesizable polymer structures. The first two, which are polyimides, imply effective gas separation performance [24].

### 4.4 RQ3: Ablation Studies and Model Analysis

**Model Components**    In light of Table 1, we analyze **three** components that impact our model's learning in various conditions. Our assessment of relative performance is based on the ratio between our method and comparative approaches. The first component is **numerical conditional encoding**. Results in Figure 4(a) highlight the superiority of clustering encoding over direct and interval-based encoding, particularly in controlling gas permeability, despite its slightly lower diversity. The second component concerns the **neural architecture for conditions**. As shown in Figure 4(b), similar to Figure 4(a), AdaLN surpasses both In-Context Conditioning and Cross-Attention in learning distribution with better condition controllability. The third component validates the importance of the **graph-dependent noise model** compared to separately applying noise to atoms and bonds. It also shows the improvement of the predictor-free Graph DiT over the predictor-guided DiGress v2, even without the graph-dependent noise model. More results on model controllability are in appendix E.

**Oracle Selections**    We analyze the robustness of Oracles in evaluating six task-related properties (three gas permeability and three small molecule properties) across six conditional generation tasks. Oracles are switched from Random Forest to Gaussian Process or Support Vector Machines for ranking generative model performance. Results in Table 3 show consistent rankings (Graph DiT, LSTM-HC, MARS, JTVAE-BO, MOOD, GDSS, GraphGA). It indicates that while perfectly approximating the truth properties of generated molecules is difficult, we could effectively compare the relative performance of various models. Graph DiT consistently ranked first among baselines.

Table 3: Oracles for Generation Evaluation: We consider three Oracles. Generative performance is ranked on average from 1 to 9 across six properties, with various Oracles yielding similar outcomes. We **highlight** models with the same ranking sequence in different Oracle evaluation.

| Avg. Rank | Random Forest | Gaussian Process | Support Vector Machine |
|---|---|---|---|
| 1 | **Graph DiT** | **Graph DiT** | **Graph DiT** |
| 2 | **LSTM-HC** | DiGress v2 | DiGress v2 |
| 3 | **MARS** | DiGress | DiGress |
| 4 | **JTVAE-BO** | LSTM-HC | LSTM-HC |
| 5 | **MOOD** | MARS | MARS |
| 6 | DiGress | JTVAE-BO | JTVAE-BO |
| 7 | DiGress v2 | MOOD | MOOD |
| 8 | **GDSS** | **GDSS** | **GDSS** |
| 9 | **Graph GA** | **Graph GA** | **Graph GA** |

## 5    Related Work

**Diffusion Models for Molecules:**    Score-based diffusion models applied noise and denoising in continuous space [33, 22]. DiGress [43] used discrete noise as transition matrices based on marginal distributions of atom and bond types. Extra predictor models are studied to guide the generation process in DiGress and GDSS [25]. Diffusion models could also be used for molecular property prediction [27], for conformation [48] and molecule generation with 3D atomic coordinates [18, 49, 3]. We focus on molecular graph generation, considering the high computational cost of accurate 3D coordinates for larger molecules like polymers [23]. We explore predictor-free diffusion guidance, instead of the classifier guidance [9, 44], for generating molecules under categorical and numerical conditions. It can be integrated with diffusion models for atomic coordinates in future research.

**Molecular Optimization:**    Optimization algorithms could optimize molecules towards property constraints, including genetic algorithms [20], Bayesian optimization [39, 50], REINFORCE [45], and reinforcement learning [30]. Both sequential and graph-based generative models [6, 21, 30], along with diverse sampling methods [47, 13], are used in conjunction with these algorithms to produce desirable molecules. These methods have been applied to both single-objective and multi-objective optimization, the latter by manually integrating multiple property conditions into a single one [5, 25]. Several challenges in molecular optimization methods remain underexplored, including the inadequate or unclear definition of multi-property relations when integration into a single objective [5], and the inaccessibility of the oracle function for property-oriented optimization during the training phase [14].

## 6    Conclusion

In this work, we solved inverse molecular design using properties as predictor-free diffusion guidance. The proposed Graph DiT performed diffusion based on the joint distribution of atoms and bonds in both forward and reverse processes. It introduced representation learning for multiple categorical and numerical properties and utilized a Transformer-based graph denoiser for conditional graph denoising. Results on multi-conditional generations and polymer inverse designs showed the remarkable generative capabilities of Graph DiT, making it suitable for designing promising molecules.

## Acknowledgements

This work was supported by NSF IIS-2142827, IIS-2146761, IIS-2234058, CBET-2332270, CBET-2102592, and ONR N00014-22-1-2507.

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

Table 4: Dataset information for all multi-conditional generation and inverse polymer design tasks. $O_2/CO_2/N_2$Perm only denotes the data statistics considering only one permeability and the generation results are presented in Table 5. The number of task conditions shown in the table does not include the timestep condition in the diffusion model.

| Datasets | # Molecule (Train/Validation/Test) | # Heavy Atom Type in Training | Min # Atoms | Max # Atoms | Avg. # Atoms | Min # Bonds | Max # Bonds | Avg. # Bonds | # Input Numerical Task Conditions | # Input Categorical Task Conditions |
|---|---|---|---|---|---|---|---|---|---|---|
| Gas Perm | 553 (331/111/111) | 11 | 3 | 48 | 27.97 | 3 | 56 | 32.67 | 5 | 0 |
| BACE | 1332 (798/267/267) | 8 | 10 | 50 | 33.67 | 10 | 54 | 36.44 | 2 | 1 |
| BBBP | 872 (522/175/175) | 9 | 3 | 50 | 24.38 | 2 | 55 | 26.26 | 2 | 1 |
| HIV | 2372 (1422/475/475) | 29 | 6 | 50 | 25.35 | 5 | 60 | 27.28 | 2 | 1 |
| $O_2/N_2$ | 609 (474/119/16) | 11 | 3 | 48 | 27.90 | 3 | 56 | 32.63 | 4 | 0 |
| $O_2$Perm only | 629 (377/126/126) | 11 | 2 | 48 | 27.42 | 3 | 56 | 32.08 | 3 | 0 |
| $CO_2$Perm only | 584 (350/177/177) | 11 | 2 | 48 | 27.59 | 3 | 56 | 32.23 | 3 | 0 |
| $N_2$Perm only | 616 (369/123/124) | 11 | 2 | 48 | 27.96 | 3 | 56 | 32.70 | 3 | 0 |

# A Details on the Denoising Model Component

## A.1 Numerical Condition Encoding

We explore several approaches for encoding numerical conditions. In addition to the clustering-based method, we consider:

1. The direct encoding approach, which employs a linear layer to map a continuous number into a high-dimensional space.

2. The interval-based approach, as described in [28], divides the label space into $N_{\text{Interval}}$ intervals. It then converts the number into an interval index, allowing us to apply one-hot encoding for the number.

## A.2 Neural Architecture for Conditions

Besides the $\text{AdaLN}$, there are two more options to integrate condition representation into molecular graph representations [34]:

1. The $\text{In-Context}$ conditioning approach adds the condition representation $\mathbf{c}$ to each row of the molecular graph representation $\mathbf{H}$ after mapping the $\mathbf{X}_G^t$ into $\mathbf{H}$ using the linear layer in the structure encoder.

2. The $\text{Cross-Attention}$ approach concatenates the timestep encoding vector with the condition representation from synthesis scores or task-related properties into a two-length sequence. In each Transformer encoder layer, this is followed by a cross-attention layer at the end of the standard multi-head self-attention layer.

# B Details on Datasets and Evaluation Methods

All experiments can be run on a single A6000 GPU card.

## B.1 Datasets and Task Conditions

As presented in Table 4, we collect popular datasets in prediction tasks for more challenging molecular generation tasks. We include a polymer dataset [40, 26] for material design. It consists of conditions of $O_2$, $CO_2$, and $N_2$, which study the numerical gas permeability for oxygen, carbon dioxide, and nitrogen, respectively. Additionally, we also study the generative performance of different models separately on the polymer data with $O_2$, $CO_2$, or $N_2$, as illustrated in Table 4. We also create three class-balanced molecule datasets from [46] for drug design: HIV, BBBP, and BACE, which study categorical properties related to the inhibition of HIV virus replication, blood-brain barrier permeability, and inhibition of human $\beta$-secretase 1, respectively. We aim to generate synthesizable

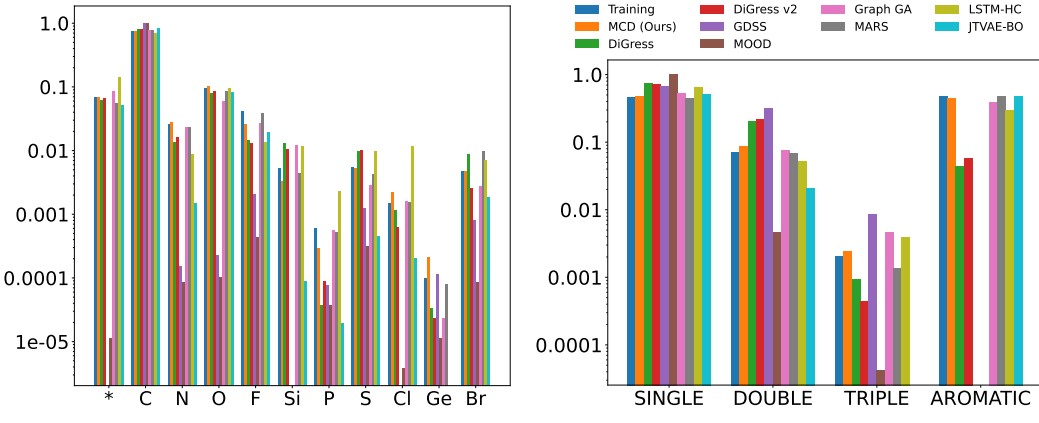

(a) Histogram for Heavy Atom Type Distribution  (b) Histogram for Bond Type Distribution

Figure 5: Histogram of Generated Distribution for Atom and Bond Types in Different Models. Results are calculated based on Table 1 for the polymer gas permeability tasks. We observe that the atom and bond type distributions from our Graph DiT's generated molecules are closer to those of the training data than other diffusion models. It indicates that Graph DiT has better capacity for learning molecular distributions.

molecules. Therefore, we add two numerical conditions for synthetic complexity scores [12, 8] for each of the tasks. For the gas separation polymer design task, we consider joint conditions for $O_2$ and $N_2$ and measure the selectivity $O_2/N_2$ as the ratio between two gas permeability scores. All polymer gas permeabilities are scaled in the **log space** following previous work [29]. We focus on experiments for polymers and molecules within 50 nodes.

## B.2   Evaluation and Metrics

We randomly split the dataset into training, validation, and testing (reference) sets in a 6:2:2 ratio. We investigate more than eight metrics to systematically evaluate the generation performance. First, we assess generation validity (Validity). Second, we evaluate the distribution learning capacity of different models by measuring heavy atom type coverage (Coverage), internal diversity among the generated examples using Tanimoto similarity (Diversity), fragment-based similarity with the reference set (Similarity), and the Fréchet ChemNet Distance with the reference set (Distance). Third, we evaluate the model's controllability by measuring the mean absolute error (MAE) between the generated condition score and the actual condition scores if the condition is numerical; otherwise, we measure the accuracy score. We follow previous work [14] to use the random forest trained on all the available data as the Oracle evaluation function. For molecular optimization algorithms, we train random forest predictors on the training set for conditional generation. For predictor-guided diffusion models, DiGress v2 and MOOD, we use the same architecture as their denoising models to train predictors for diffusion guidance. Given the conditions in the test set, we report the generation performance by generating 10,000 examples for the six multi-conditional generation tasks and 1,000 examples for the polymer inverse design problem focused on selectivity.

## B.3   Datasets and Tasks in Figure 1

Using the same dataset from the $O_2/N_2$ polymer inverse design task, we keep 100 polymers to provide condition sets for testing and split the rest into training and validation sets in a 0.65:0.35 ratio. We apply our proposed Graph DiT for both single-conditional and multi-conditional approaches, focusing on three properties: (1) Synth. score for synthesizability [12], (2) $O_2$ permeability, and (3) $N_2$ permeability. The single-conditional approach generates 30 polymers per condition for each test data point, totaling 9,000 polymers. In contrast, the multi-conditional approach generates 30 polymers for each set of conditions per test data point, resulting in 3,000 polymers. We rank these polymers based on the mean absolute error between the generated properties (evaluated by a random forest model trained on all the data to simulate the Oracle function) and the conditional property. For each test data point, we also rank the best multi-conditional polymer in different single-conditional

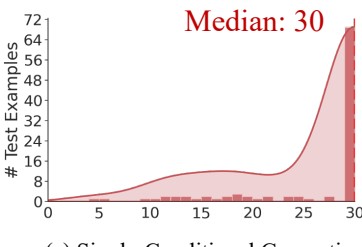 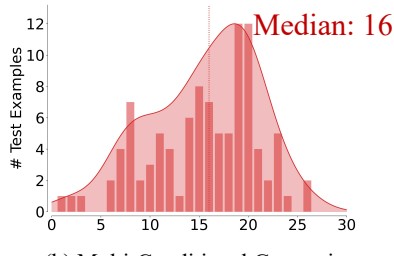

(a) Single-Conditional Generation       (b) Multi-Conditional Generation

Figure 6: We compare the average ranking of generated polymers with desirable properties. First, we generate three sets of polymers using a single-conditional approach for each condition. In (a), as shown in Figure 1, we find the ranking of the shared structure for each multi-condition requirement. In (b), polymers are generated using a multi-conditional approach, and for each, we identify the highest ranking among the three single-conditional sets. Then, we calculate the median of these maximum ranking positions, which is 16, approximately $2\times$ better than single-conditional generation, which has a median value greater than 30.

sets. For the single-conditional approach, we identify a common polymer meeting various properties and visualize the minimum top $K$ value distribution across all 100 test points.

In addition to Figure 1, we compute the median rankings of the multi-conditionally generated polymers within the single-conditional sets. The results are shown in Figure 6. Both Figure 1 and Figure 6 demonstrate the advantages of multi-conditional generation over single-conditional generation.

## C   Details on Multi-Conditional Generation Results

We show results for three polymer generation tasks in Table 1 and molecule generation tasks in Table 2. As complementary results for Table 1, we present new results on generation using only one gas permeability in Table 5. We also compare the distributions of atom and bond types between generated and training data in Figure 5. Furthermore, Figure 7 visualizes the two-dimensional molecular data distribution of both training and generated molecules across various generative models.

### C.1   Discussion on Diffusion Model Baselines

While diffusion models like GDSS [22] and DiGress [43] show promise in unconditional tasks, their performance in multi-conditional generations needs improvement for fitting training distributions and achieving more controllable results. As indicated by Figure 5(a), the generation of GDSS [22] often collapses to carbon elements with Gaussian noise in the continuous diffusion state-space. MOOD [25] improves atom type coverage by adding predictor guidance and an out-of-distribution hyper-parameter, but it is hard to fit the training distribution, as visualized in Figure 7(h). DiGress and its predictor-guided variant [43] (i.e., DiGress v2), using discrete state-space and transition matrices for diffusion noise, outperform GDSS and MOOD in distribution fitting and internal diversity in Tables 1, 2 and 5. However, as indicated in Figure 7(e) and Figure 7(f), these two models still generate too many out-of-distribution examples without justification of the generalization capacity. While GDSS and MOOD show lower average MAE in polymer conditional generation tasks, their subpar distribution learning performance and the results from Table 2 suggest that this may be due to the carbon element, which may be a confounder and affect the evaluation of the correlation between the model and controllability in polymer tasks.

### C.2   Discussion on Molecular Optimization Baselines

Popular molecular optimization baselines are competitive in molecular generation tasks. Earlier studies have noted their strong performance: Gao et al. [14] showed their effectiveness in the standard molecular optimizations with a combined optimization target, and Tripp and Hernández-Lobato [41] found that genetic algorithm often outperforms recent methods in unconditional generation. We

Table 5: Generation of 10K Polymers: Results on a numerical synthesizability score (Synth.) and a numerical properties (gas permeability for $O_2$, $N_2$, or $CO_2$). MAE is calculated between input conditions and generated properties. Best results are **highlighted**.

| Tasks | Model | Validity ↑ (w/o rule checking) | Coverage ↑ | Diversity ↑ | Similarity ↑ | Distance ↓ | Synth. ↓ | Property ↓ | Avg. ↓ |
|---|---|---|---|---|---|---|---|---|---|
| | | | | Distribution Learning | | | Condition Control | | |
| Synth. & $O_2$ Perm | Graph GA | 1.0000 (N.A.) | 11/11 | 0.8885 | 0.9180 | 8.3925 | 1.3254 | 1.8962 | 4.4521 |
| | MARS | 1.0000 (N.A.) | 10/11 | 0.2263 | 0.5170 | 26.6354 | **0.8502** | 1.8853 | 3.6472 |
| | LSTM-HC | 0.9896 (N.A.) | 10/11 | 0.8898 | 0.8015 | 17.5424 | 1.2727 | 1.1323 | 3.8278 |
| | JTVAE-BO | 1.0000 (N.A.) | 8/11 | 0.7672 | 0.8895 | 21.3698 | 0.9703 | 1.3257 | 3.2971 |
| | DiGress | 0.9934 (0.3756) | 11/11 | 0.9156 | 0.2648 | 19.9364 | 2.5093 | 1.6424 | 5.2492 |
| | DiGress v2 | 0.9842 (0.4237) | 11/11 | **0.9204** | 0.2311 | 20.4500 | 2.3444 | 1.6445 | 5.1255 |
| | GDSS | 0.9910 (0.4482) | 1/11 | 0.8891 | 0.0058 | 36.5735 | 1.6074 | 1.4803 | 4.3219 |
| | MOOD | 0.9952 (0.4764) | 9/11 | 0.8898 | 0.0072 | 36.0428 | 1.5089 | 1.4595 | 4.2277 |
| | Graph DiT-LC (Ous) | 0.9826 (0.8974) | 11/11 | 0.8941 | 0.9662 | 5.8940 | 1.1302 | 0.8341 | 3.1345 |
| | Graph DiT (Ours) | 0.8242 (0.8974) | 11/11 | 0.8788 | **0.9688** | **5.4287** | 1.0672 | **0.7843** | **2.9442** |
| Synth. & $N_2$ Perm | Graph GA | 1.0000 (N.A.) | 11/11 | 0.8806 | 0.8760 | 9.4945 | 1.2593 | 2.3122 | 4.7050 |
| | MARS | 1.0000 (N.A.) | 10/11 | 0.1118 | 0.3269 | 33.6684 | 2.1089 | 2.3316 | 5.8333 |
| | LSTM-HC | 0.9901 (N.A.) | 10/11 | 0.8924 | 0.7804 | 17.6290 | 1.4530 | 1.2798 | 4.1070 |
| | JTVAE-BO | 1.0000 (N.A.) | 10/11 | 0.7741 | 0.7264 | 22.5093 | **0.9414** | 1.2874 | 3.2911 |
| | DiGress | 0.9798 (0.3326) | 11/11 | 0.9150 | 0.2670 | 21.1077 | 2.7562 | 2.0242 | 5.9103 |
| | DiGress v2 | 0.9801 (0.3759) | 11/11 | **0.9180** | 0.1842 | 20.7820 | 2.4734 | 1.9538 | 5.5614 |
| | GDSS | 0.9941 (0.8000) | 3/11 | 0.8543 | 0.0030 | 33.3815 | 1.5277 | 1.5886 | 4.3365 |
| | MOOD | 0.9980 (0.4453) | 11/11 | 0.8857 | 0.0028 | 34.9385 | 1.5087 | 1.7018 | 4.3261 |
| | Graph DiT-LC (Ous) | 0.9803 (0.9054) | 11/11 | 0.8894 | 0.9670 | 5.9049 | 1.1908 | 0.9721 | 3.3105 |
| | Graph DiT (Ours) | 0.8165 (0.9054) | 11/11 | 0.8726 | **0.9713** | **5.7943** | 1.0969 | **0.9472** | **3.1124** |
| Synth. & $CO_2$ Perm | Graph GA | 1.0000 (N.A.) | 11/11 | 0.8889 | 0.9134 | 7.1234 | 1.3427 | 1.8548 | 4.4079 |
| | MARS | 1.0000 (N.A.) | 11/11 | 0.8460 | 0.9083 | 8.9201 | 1.1623 | 1.4808 | 3.8612 |
| | LSTM-HC | 0.9893 (N.A.) | 10/11 | 0.8938 | 0.7262 | 16.1368 | 1.4018 | 1.1436 | 3.8079 |
| | JTVAE-BO | 1.0000 (N.A.) | 7/11 | 0.7671 | 0.7978 | 22.9047 | **1.0550** | 1.1663 | 3.2622 |
| | DiGress | 0.9802 (0.3741) | 11/11 | 0.9100 | 0.1576 | 19.6117 | 2.4554 | 1.5377 | 5.1926 |
| | DiGress v2 | 0.9868 (0.2486) | 11/11 | **0.9137** | 0.2686 | 20.1563 | 2.8087 | 1.5590 | 5.5939 |
| | GDSS | 0.9876 (0.6987) | 1/11 | 0.8786 | 0.0026 | 32.1841 | 1.4679 | 1.3584 | 4.0440 |
| | MOOD | 0.9881 (0.7880) | 11/11 | 0.8690 | 0.0025 | 30.9310 | 1.5463 | 1.3443 | 4.1464 |
| | Graph DiT-LC (Ours) | 0.9836 (0.8841) | 11/11 | 0.8916 | 0.9247 | 5.7776 | 1.2991 | 0.8603 | 3.3394 |
| | Graph DiT (Ours) | 0.8291 (0.8841) | 11/11 | 0.8743 | **0.9403** | **5.6815** | 1.2225 | **0.7728** | **3.1155** |

Table 6: Complete results on 1,000 generated polymers for the inverse $O_2/N_2$ gas separation polymer design. # UB is the count of generated polymers successfully identified (by Oracle functions) as upper bound instances defined by Robeson [38].

| Model | Validity ↑ w/o rule checking | Coverage ↑ | Diversity ↑ | Similarity ↑ | Distance ↓ | Synth. ↓ | $O_2$ ↓ | $N_2$ ↓ | Avg. MAE ↓ | # UB ↑ |
|---|---|---|---|---|---|---|---|---|---|---|
| | | | Distribution Learning | | | Condition Control | | | | |
| Graph GA | 1.0000 (N.A.) | 10/11 | 0.8848 | 0.3734 | 29.9060 | 1.6545 | 1.8720 | 2.1984 | 8.7726 | 57 |
| MARS | 1.0000 (N.A.) | 11/11 | 0.7886 | 0.1922 | 32.5679 | 1.4909 | 1.7940 | 2.2170 | 9.3112 | 51 |
| LSTM | 0.9910 (N.A.) | 10/11 | 0.8940 | 0.1758 | 37.1556 | **1.3553** | 1.5066 | 1.8791 | 10.5529 | 45 |
| JTVAE-BO | 1.0000 (N.A.) | 7/11 | 0.7849 | 0.2541 | 33.6430 | 2.0723 | 1.7653 | 2.1998 | 9.7722 | 33 |
| DiGress | 0.9930 (0.3120) | 9/11 | 0.9019 | 0.1156 | 30.5716 | 1.6892 | 1.1680 | 1.3329 | 8.6061 | 69 |
| DiGress v2 | 0.9940 (0.1760) | 11/11 | **0.9075** | 0.2793 | 29.6239 | 2.1370 | 1.1847 | 1.3743 | 8.4707 | 85 |
| GDSS | 0.9910 (0.9180) | 1/11 | 0.8210 | 0.0000 | 41.7499 | 2.3508 | 1.4097 | 1.8328 | 11.7813 | 56 |
| MOOD | 0.9960 (0.5640) | 9/11 | 0.8803 | 0.0000 | 46.6095 | 1.5963 | 1.3921 | 1.7360 | 12.8551 | 81 |
| Graph DiT-LC | 0.975 (0.7170) | 10/11 | 0.8966 | **0.6401** | 26.1647 | 1.5119 | 0.8388 | 0.9035 | 3.2541 | **90** |
| Graph DiT (Ours) | 0.7800 (0.7170) | 11/11 | 0.8838 | 0.6028 | 26.3378 | 1.4081 | **0.7476** | **0.8213** | **2.9770** | 68 |

observe their competitive performance in multi-conditional settings, characterized by high validity, good atom type coverage, and distribution similarity in Tables 1, 2 and 5. MARS [47] and JTVAE [21] perform well for controlling the synthetic accessibility score [12] but are less effective at controlling specific task properties like gas permeability, BACE, BBBP, and HIV. For example, in generation tasks with the categorical task condition, the generated examples only achieve around 50% accuracy in hitting the input condition.

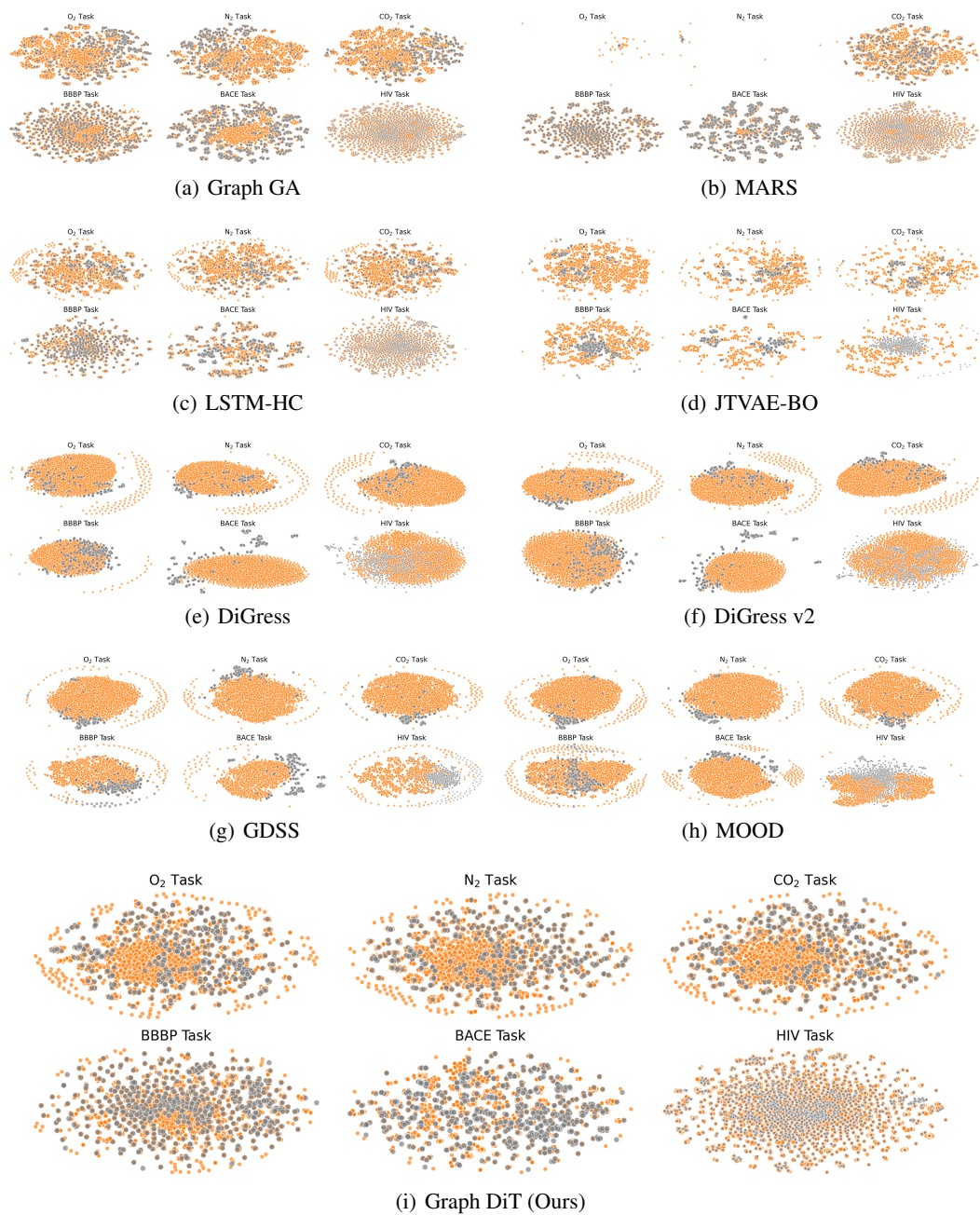

Figure 7: Distribution of training (grey-colored) and generated (orange-colored) molecules. The generated distribution in Figure 7(i) is from Graph DiT, and the visualization shows that the generated molecular data points fit the training distribution well, with reasonable interpolation and extrapolation in the training data space.

## C.3 Discussion on Training Dynamics

In Figure 8, we illustrate changes in various indicators on the validation set during model training. We note an increase in generated validity and similarity to the validation reference set, along with a decrease in distance to the reference set and errors between generated properties and conditions, indicating gradual improvement in conditional generation over epochs. However, a trade-off between distribution fitting and internal diversity is observed in our current model, suggesting that further work on enhancing generation diversity could be promising.

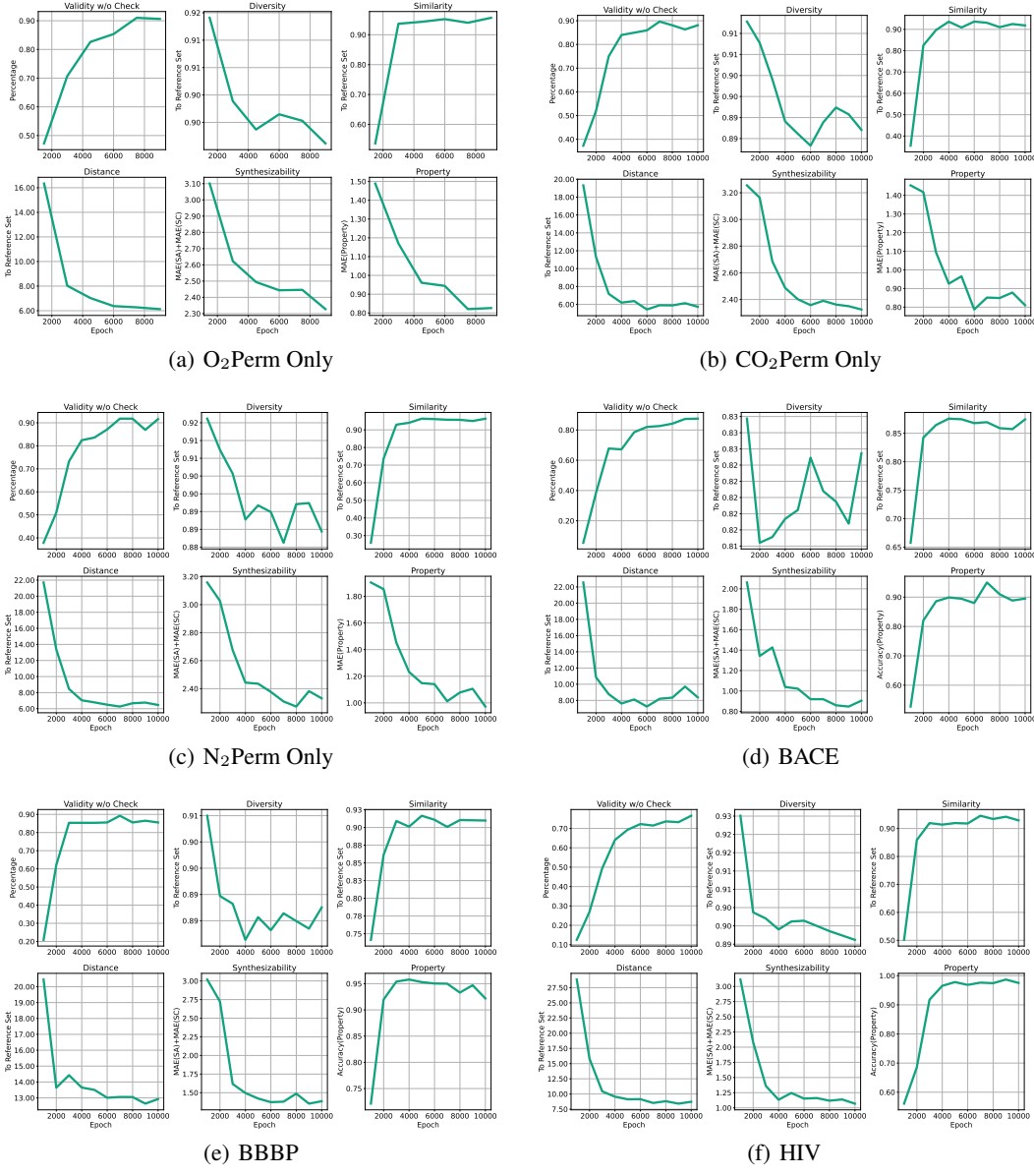

Figure 8: Change of indicators on the validation set during model training. The generated validity and similarity to the validation reference set have increased, accompanied by a decrease in distance to the reference set and errors between generated properties and conditions. We also observe a trade-off between distribution fitting and internal diversity over epochs.

## C.4 Discussion on Uniqueness, and Novelty

We evaluate the model's performance on Novelty and Uniqueness. Unlike unconditional generation, multi-conditional generation involves generating multiple possible molecules under the same condition. Therefore, we compute these metrics across different sets of conditions rather than for generated molecules under the same conditions. Results are presented in Table 7.

Graph DiT demonstrates reasonable performance on these metrics. However, higher Novelty and Uniqueness values do not necessarily indicate better performance, as they may not reflect the model's ability to design satisfactory molecules with desirable properties. Moreover, these values risk leading to misleading conclusions. For instance, AddCarbon achieves nearly perfect scores (99.94% Novelty

Table 7: Comparison of Novelty and Uniqueness across different conditions

| Metric | Graph GA | MARS | LSTM-HC | JTVAE-BO | Digress | DiGress v2 | GDSS | MOOD | Graph DiT |
|---|---|---|---|---|---|---|---|---|---|
| Novelty | 0.9950 | 1.0000 | 0.9507 | 1.0000 | 0.9908 | 0.9799 | 0.9190 | 0.9867 | 0.9702 |
| Uniqueness | 1.0000 | 0.7500 | 0.9550 | 0.6757 | 1.0000 | 0.9730 | 0.1622 | 0.9820 | 0.8829 |

and 99.86% Uniqueness) according to [37, 41], yet it randomly adds carbon atoms to existing molecules, resulting in new molecules that are not practically useful [37].

# D  Details on Polymer Inverse Design

We aim to design polymers with high $O_2$ and low $N_2$ permeability, showing refined control of models over these properties. This is reflected in the selectivity, defined as the $O_2/N_2$ permeability ratio. Robeson [38] has identified an inherent trade-off between gas permeability and selectivity, known as the upper-bound. Ideally, high-performance polymers should fall in the above-the-bound region, demonstrating an effective combination of permeability and selectivity. We have 609 polymers with annotated permeability values for both gases. 16 above-the-bound polymers are included in the test/reference set and excluded from the training set. We generate 1,000 polymers conditional on test set labels.

## D.1  Survey Setup on Generated Polymers

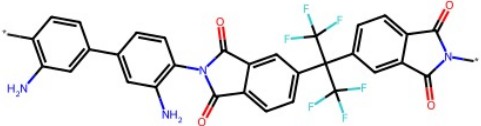

Figure 9: The reference polymer structure in the case study has conditions {SAS=3.8, SCS=4.3, $O_2$Perm=34.0, $N_2$Perm=5.2}.

We aim to gather expert evaluations on the generation performance of various methods. We conduct a study using a test data point from the $O_2/N_2$ gas separation inverse design task, taking its properties as conditions. The structure of the selected data point is presented in Figure 9. We display 25 generated polymers, each with its properties, alongside three real polymers from the training dataset as references. The first real polymer serves as the test reference, while the other two, similar in properties to the first, also aid experts in assessing the generated polymers. The properties of these generated polymers are predicted using a well-trained random forest model. Experts are asked to rank the generated polymers from 1 to 25, considering: (1) Structures of real polymers with desirable properties; (2) Predicted properties of generated polymers, displayed beneath each visualization. Here, a rank of 1 represents the best example as per domain knowledge, while 25 is the least favorable. Rankings are then converted to utility scores (UtS) ranging from 0 to 1 using $\frac{1}{\text{ranking}}$, allowing us to quantify the relative performance of different generation methods. The agreement score (AS) could be obtained by $\exp\left(-25 \times \text{Variance}(\text{UtS}_1, \text{UtS}_2, \text{UtS}_3, \text{UtS}_4)\right)$, where $\text{UtS}_i$ denotes the utility score from the $i$-th domain expert. (3) Finally, we select top-3 polymers for each generative models and present them in Figure 3.

## D.2  Results on Inverse Design

We present the inverse design results of all 1,000 generated polymers in validity, distribution learning, and condition control in Table 6. **# UB** is the count of generated polymers successfully identified as upper bound instances. Higher # UB indicates that Graph DiT has a higher likelihood of generating candidates for excellent $O_2$ and $N_2$ gas separation. The smallest MAE across most properties and a 9.9% average MAE improvement over baselines highlight Graph DiT's superior control in generating examples closely aligned with multiple conditions.

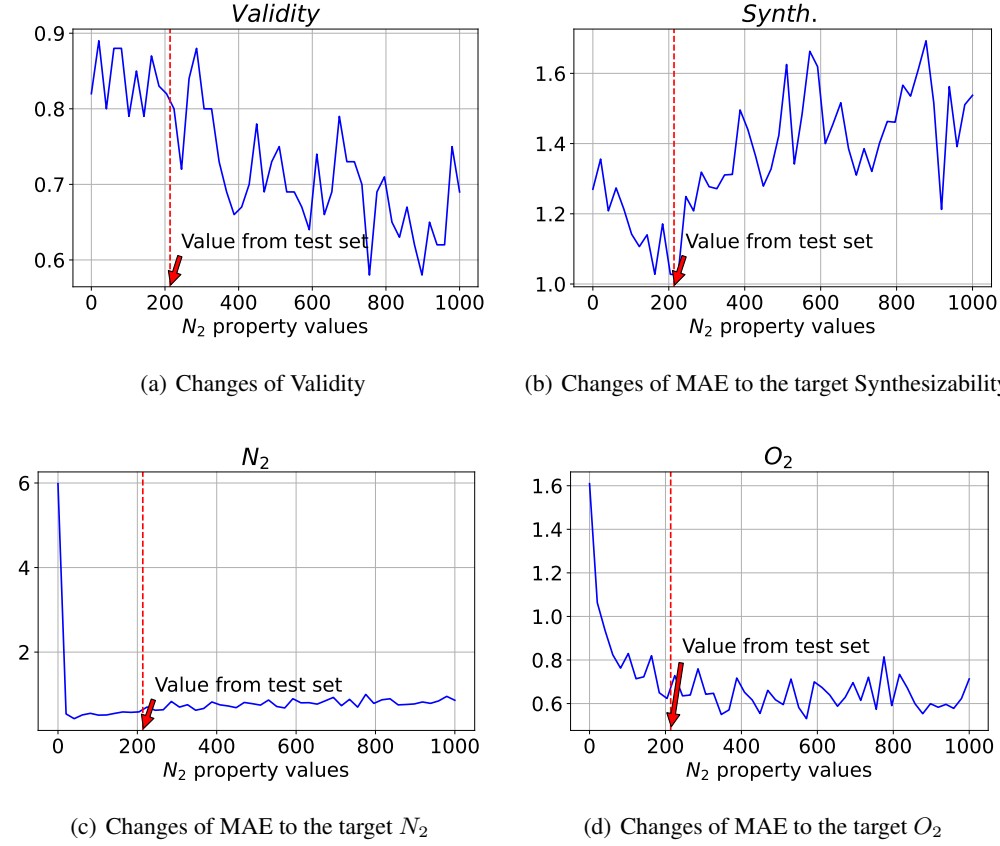

(a) Changes of Validity

(b) Changes of MAE to the target Synthesizability

(c) Changes of MAE to the target $N_2$

(d) Changes of MAE to the target $O_2$

Figure 10: Analysis of Model Controllability when Varying $N_2$ Values: The true $N_2$ value from the test set is 213.75. We note that the controllability performance (i.e., MAE value) for $N_2$ and $O_2$ is measured in log space.

# E  Details and More Results on Model Analysis

## E.1  Case Studies on Generation Controllability

We conduct a new case study on the $O_2/N_2$ polymer dataset, studying the controllability on three properties synthesizability score, $O_2$ and $N_2$ properties with varied $N_2$ property values. We select a polymer example from the test set and vary its $N_2$ while keeping other properties fixed. The $N_2$ property from the test polymer is 213.75, and we vary it from 0 to 1000. We sample 50 values within this range and generate 100 polymer graphs conditioned on multiple properties with each sampled value. We evaluate various metrics, including the chemical validity of the generated polymers.

We visualize results in Figure 10. We consistently observe that validity and controllability performance improve as the values approach 213.75, derived from a real test polymer. Conversely, performance deteriorates when the sampled $N_2$ values are closer to the extremes of the sampling range (0 or 1000). This observation underscores the interdependency between conditions, where less frequent combinations of different properties may be more challenging to learn. Moreover, the model performs well across a elatively large range from 0 to 1000 in terms of validity, $O_2$, and synthesizability score control. This demonstrates good generalization of the proposed method in capturing complex condition interdependencies.

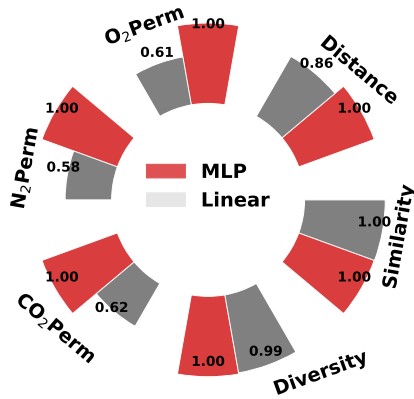

Figure 11: Ablation studies on the final MLP layer

Table 8: Training Performance of Oracle Methods: we train the models on all polymers or small molecules in a task to simulate the Oracle. Results from the random forest model are **highlighted** because it has the lowest training MAE and highest training AUC.

| | $O_2Perm$ (MAE) | $N_2Perm$ (MAE) | $CO_2Perm$ (MAE) | BACE (AUC) | BBBP (AUC) | HIV (AUC) |
|---|---|---|---|---|---|---|
| Random Forest | **0.3662** | **0.4006** | **0.3486** | **0.9895** | **0.9954** | **0.9996** |
| Gaussian Process | 1.9631 | 2.3806 | 1.8543 | 0.9610 | 0.9943 | 0.9511 |
| Support Vector Machine | 0.7462 | 0.9509 | 0.8594 | 0.8889 | 0.9472 | 0.9304 |

### E.2 Ablation Studies on Final MLP

In addition to the three components related to conditioning effectiveness of Graph DiT studied in Section 4.4, we also examine the importance of the final layer MLP for conditional graph denoising. Results in Figure 11 show that MLP significantly outperforms a linear layer [34].

### E.3 Details on Oracle Simulation

We train three types of Oracles based on Random Forest, Gaussian Process, and Support Vector Machine on all polymers or molecules in a task to evaluate the properties of generated polymers conditional on $O_2Perm$ only, $N_2Perm$ only, $CO_2Perm$ only, BACE, BBBP, or HIV. The training performance (MAE or AUC) is presented in Table 8. Our findings show that the random forest achieves the lowest MAE and highest AUC scores, leading us to select it for simulating oracles in our generation evaluation process.

