# OpenReview forum: "Graph Diffusion Transformers for Multi-Conditional Molecular Generation"
_NeurIPS.cc/2024/Conference — NeurIPS 2024 oral_

### Official Review · Reviewer_rG3q · 2024-06-24

**Soundness:** 3
**Presentation:** 4
**Contribution:** 3
**Rating:** 6
**Confidence:** 3

**Summary:**

In molecular generation tasks, there is often a desire for the generator to produce molecules that satisfy multiple properties simultaneously. Previous architectural designs did not pay particular attention to the scenario of optimizing multiple constraints in tandem. Therefore, the author has proposed a Multi-Conditional Graph Diffusion Transformer. In the noise addition phase, the author has employed a novel noise addition method, successfully establishing connections between the graph's nodes and edges. At the same time, the author has designed a new condition encoder and a graph diffusion transformer architecture. This architecture is capable of addressing both numerical and categorical conditions simultaneously. Experiments have demonstrated that the proposed method exhibits excellent performance.

**Strengths:**

- The writing is mostly clear and easy to follow
- The evaluation is comprehensive.

**Weaknesses:**

- some comparisons are not fair, see the questions for detail.
- some metrics of evaluation is not so suitable, see questions for detail

**Questions:**

**Major Issues**

- The comparison presented in Figure 1 is not fair. Molecules that rank low on a certain single property may not necessarily rank low on other properties. Therefore, for a fair comparison, the statistical graph shown in Figure 1(b) should follow the same format as in Figure 1(a). That is, the authors should calculate and visualize the minimal $ K $ at which a molecule can satisfy all the constraints within rank $ K $ for all the generated molecules.
- All the datasets used for drug design suffer from a severe class imbalance issue, i.e., the ratio of positive to negative samples is less than 1 to 20. Could the authors specify the approximate ratio of positive to negative samples in the input condition (label) when calculating the accuracy in Table 2? If the ratio of positive to negative samples follows the original dataset's proportion, using ROC-AUC seems to be a more reasonable metric.
- How is the Avg. Rank calculated in Table 2?
- In section 3.1, if the setting of $\mathbf{Q}\_V$ follows the setting of Digress, and $\mathbf{Q}\_{EV}$ is derived from the marginal distribution of $V$, then we have, for any $k$, $\sum\_{i=1}^{F_V}\mathbf{P}\_{k,i}=N+1 \neq 1$, where $\mathbf{P}=\mathbf{X}\_G^{t-1}\mathbf{Q}\_G^t$, $\mathbf{P}\_{k,i}$ represents the probability of node $k$ being of category $i$. The edges have the same issue. Therefore, $q(\mathbf{X}\_G^t|\mathbf{X}_{G}^{t-1})$, as defined by the formula in the article, does not seem to be a distribution. However, in the specific implementation process, the graph is sampled from the "distribution" instead of directly using the probability as the edge and node features, which means the flaws mentioned above  theoretically should not affect the model's performance. But the authors need to make corrections in the writing.

**Minor Issues**

- Many diffusion-based generation methods for conditional generation only require the introduction of conditions during the generation process, without the need to incorporate conditions during the training process. Can the proposed method achieve the same?


**If you can address all the major issues, I would be very willing to raise my score**

**Limitations:**

I believe the societal impacts and limitations of this work are discussed to a sufficient extent.

---

> ### Author Rebuttal · Authors · 2024-08-06
>
> # Major 1: Ranking comparison in Figure 1
>
> ## Results are updated to show ~$2\times$ improvement
>
> Thank you for your suggestion. We have created a new figure (attached in the rebuttal PDF) based on your feedback. We first identify the maximum ranking position among the three single-conditional generation sets for each multi-condition generated polymer. Then, we calculate the median of these maximum ranking positions, which is 16—approximately $2\times$ better than single-conditional generation, which has a median value greater than 30.
>
> # Major 2: Data imbalance issue
>
> ## The datasets are balanced, and the accuracy is reasonable
>
> Thank you for your comment. Please refer to Line 200:
> > For drug design, we create three class-balanced datasets from MoleculeNet [45]: HIV, BBBP, and BACE...
>
> The positive-to-negative ratio is 1:1, making accuracy a reasonable metric in this case.
>
> # Major 4: How avg. rank calculated
>
> It is calculated by averaging the ranking positions of model performance for each condition (property).
>
> # Major 5: The issue with $q(\mathbf{X}_G^{t} | \mathbf{X}_G^{t-1})$
>
> ## We correct this issue
>
> Thank you for pointing this out! The term $\mathbf{X}_G^{t-1}\mathbf{Q}_G^{t}$ results in unnormalized probabilities. In the implementation, we need to split it into node and edge components and normalize them separately. We have corrected this issue and revised the description of Eq. (6) as follows:
>
> We introduce a new diffusion noise model. At each forward diffusion step $t$, noise is applied to $\mathbf{X}_G^{t-1}$, resulting in an unnormalized probability $\mathbf{\tilde{p}}= \mathbf{X}_G^{t-1} \mathbf{Q}_G^t$. We first separate and normalize the first $F_V$ columns of $\mathbf{\tilde{p}}$ to obtain the noisy node states $\mathbf{X}_V^t$. We then reshape and normalize the remaining $N \cdot E$ dimensions to obtain the edge states $\mathbf{X}_E^{t}$. These components are combined to construct $\mathbf{X}_G^{t}$.
>
> # Minor 1: Can the proposed method achieve no condition during training
>
> Yes, here are two strategies:
>
> 1. Replace the condition encoder with a molecular encoder that uses a self-supervised task during large-scale pretraining. In this case, the generation conditions should be the molecular structure rather than labels. For property conditions, we can retrieve molecular structures with similar labels.
> 2. Learn null embeddings of the conditions during pretraining. Although fine-tuning the condition encoder for specific tasks will still be necessary, pretraining can help reduce costs and label requirements.
>
> All these strategies are promising directions for extending Graph DiT.

---

> > ### Comment · Reviewer_rG3q · 2024-08-07
> > **Further Questions about Major 2**
> >
> > Am I correct in assuming that the dataset employed in your study is a balanced subset of the HIV dataset, which includes an equal number of positive and negative samples, rather than the complete dataset?

---

> > > ### Author Response · Authors · 2024-08-07
> > > **Further response about Major 2**
> > >
> > > Thank you for your prompt feedback. Yes, it is the balanced subset with 2,372 examples. All the information about the dataset is provided in Table 4 (Appendix).

---

> > > > ### Comment · Reviewer_rG3q · 2024-08-07
> > > > **Final Response**
> > > >
> > > > Your response has addressed all my concerns. I would like to increase my rating to 6.

---

> > > > > ### Author Response · Authors · 2024-08-07
> > > > > **Thank you for raising the score**
> > > > >
> > > > > We are encouraged that your concerns have been addressed. Thank you again for your valuable and helpful feedback! We welcome any new discussion.

---

### Official Review · Reviewer_N56G · 2024-07-02

**Soundness:** 3
**Presentation:** 2
**Contribution:** 3
**Rating:** 5
**Confidence:** 4

**Summary:**

This paper presents Graph DiT, which initially learns conditions (including categorical and numerical properties) through clustering and one-hot encodings. Subsequently, Graph DiT utilizes a Transformer architecture during the diffusion denoising phase to refine noisy molecular graphs incorporating these conditions. Experimental results underscore Graph DiT's efficacy in multi-conditional generation and polymer inverse design, emphasizing its capability to innovate in molecule creation.

**Strengths:**

- **This paper makes somewhat novel contributions**: For example, unlike previous studies that treat node and edge state transitions independently, potentially misaligning with the denoising process, this research proposes a graph-dependent noise model. Graph DiT constructs a transition matrix based on the joint distribution of nodes and edges, enhancing the coherence of the denoising process.

- **The experiments address three primary questions**: (1) The authors validated the generative capabilities of Graph DiT against baseline models from molecular optimization and diffusion. (2) The authors explored polymer inverse design for gas separation. (3) They conducted additional analysis to further examine the capabilities of Graph DiT.

- A portion of the experimental results showed superior performance compared to the baselines.

**Weaknesses:**

- The paper should emphasize the main contributions and acknowledge its limitations.

- The presentation of the paper should be improved as it is difficult to follow.

- **Lack of experiments**: Similar to DiGress, the paper should compute statistics such as uniqueness, novelty, and VUN (Valid, Unique, and Novel graphs) to provide a comprehensive assessment of the method's efficacy and innovation in graph generation and optimization. These metrics are crucial for evaluating the quality and originality of the generated graphs, ensuring a thorough comparison with existing approaches in the field.

- The experimental results in Table 2 did not demonstrate superior performance compared to other baseline methods. The effectiveness of the proposed Graph DiT appears somewhat trivial to some extent. Only the results in conditional control show good performance.

**Questions:**

- Please refer to the "weaknesses" section above.

- The authors argue in lines 25 to 27 that the diverse scales and units of properties present a significant challenge in multi-property optimization. This diversity can complicate the comparison and combination of different properties, potentially leading to skewed optimization results. In my view, a straightforward solution to this issue could involve scaling the properties to a common range, such as 0 to 1. This approach would normalize the scales and facilitate fair comparisons and effective optimization across different properties, thereby addressing the challenge effectively.

- What does "LCC" signify in DiT-LCC?

- What visualization method is used in Figure 6? Is it a linear technique like PCA, or a non-linear method such as UMAP, utilized for dimensionality reduction?

---

> ### Author Rebuttal · Authors · 2024-08-06
>
> # W1, W2: Contribution, limitation, and paper presentation
>
> Thank you for your comment. We can highlight lines 41-43 and 62-64 using italics to emphasize the contributions and organize them into bullet points. Any further suggestions or discussion are highly appreciated.
>
> We discussed limitations in Section 4.4 (Lines 281, 288-289) regarding generation diversity and Oracle functions used for evaluation. We revise the main paper to acknowledge these again in the Conclusion.
>
> We have corrected Figure (4) colors and revised Eq. (6) for clarity. Any further suggestions or discussions are welcome.
>
> # W2: Lack of metrics (Novelty & Uniqueness)
>
> ## Results on required metrics
>
> We show that Graph DiT achieves good values for novelty and uniqueness.
>
> | Method     | Validity | Novelty | Uniqueness | V * N * U |
> |------------|----------|---------|------------|---------------------------------|
> | Graph GA   |   1.0000 |  0.9950 |     1.0000 |                          0.9950 |
> | MARS       |   1.0000 |  1.0000 |     0.7500 |                          0.7500 |
> | LSTM-HC    |   0.9910 |  0.9507 |     0.9550 |                          0.8997 |
> | JTVAE-BO   |   1.0000 |  1.0000 |     0.6847 |                          0.6847 |
> | Digress    |   0.9913 |  0.9908 |     0.9730 |                          0.9556 |
> | DiGress v2 |   0.9812 |  0.9799 |     0.9820 |                          0.9442 |
> | GDSS       |   0.9190 |  0.9190 |     0.1532 |                          0.1293 |
> | MOOD       |   0.9867 |  0.9867 |     0.9730 |                          0.9473 |
> | Graph DiT  |   0.9760 |  0.9702 |     0.8919 |                          0.8445 |
>
> ## Flaws in uniqueness and novelty
>
> We did not choose uniqueness and novelty as major metrics because recent research shows they can be flawed [1]: randomly adding carbons to existing molecules can yield almost 100% novelty and uniqueness. Instead, we use internal diversity to better reflect generation diversity.
>
> ## Current evaluation has 9 metrics, enhanced by expertise
>
> We defend our nine metrics for evaluating molecular quality, practical utility, and diversity. These metrics cover chemical validity, distribution matching (diversity, distance, similarity), and multi-condition controllability.
>
> Additionally, Section 4.3 includes case studies with domain experts, providing valuable assessments often lacking in previous studies.
>
> Based on the above points, we respectfully request a reconsideration of the assessment if the reviewer finds these metrics sufficient. We welcome further discussion on this matter.
>
> # W3: Results in Table 2
>
> ## From 0.6 to 0.9, Graph DiT significantly outperforms the baselines
>
> Thank you for your comment. Condition control is a core goal in inverse molecular design, as we need to design drugs/materials that meet human requirements. As shown in Table 2, Graph DiT effectively improves existing baselines from 0.6 to 0.9 in this regard.
>
> Based on the above points, we respectfully request a reconsideration of the assessment that "... results in Table 2 did not demonstrate superior performance..."
>
> ## More thoughts on effectiveness and practical utility
>
> We value the reviewer's opinion that an ideal model should excel across all metrics. However, we respectfully argue that defining ground-truth diversity and distribution distance is still debated with many choices (e.g. uniqueness or internal diversity). Therefore, we use up to 9 metrics to comprehensively evaluate model performance from diverse perspectives. Comparing very similar numbers, such as a distance metric of 6.7 vs. 7.0, is less compelling for demonstrating a model's ability to generate practically useful molecules. Significant improvement in multi-conditional controllability is more indicative of practical utility. Additionally, in Section 4.3, we involve domain experts to verify the generated results in real applications.
>
> We greatly appreciate the reviewer’s comments and invite further discussion on the matter for any remaining concerns.
>
> # Q2: Diverse scales of properties
>
> ## Lines 217-218 align with the reviewer’s question
>
> Thank you for the excellent question. As detailed in Lines 217-218, our implementation for molecular optimization baselines aligns with your question. Results in Tables 1 and 2 demonstrate that Graph DiT effectively outperforms these baselines.
>
> # Q3 LCC meanings
>
> ## Lines 185-188: LCC denotes largest connected component
>
> Thank you for your question. Please see Lines 185-188 for more:
> > A common way of converting generated graphs to molecules selects only the largest connected component [42], denoted as Graph DiT-LCC in our model. For Graph DiT, we connect all components by randomly selecting atoms. It minimally alters the generated structure to more accurately reflect model performance than Graph DiT-LCC.
> >
>
> # Q4: Visualization method in Figure 6
>
> We use PCA (Principal Component Analysis) to reduce the dimensionality of Morgan Fingerprints [2] to two dimensions for visualization.
>
> # Reference
>
> [1] Genetic Algorithms are Strong Baselines for Molecule Generation. 2023.
>
> [2] Extended-Connectivity Fingerprints. JCIM 2010.

---

> ### Comment · Reviewer_N56G · 2024-08-12
> **Response to Authors' Rebuttal**
>
> Thank you for the authors' responses. The authors' rebuttal does not convince me.
>
> Firstly, I disagree with the authors' claim that "randomly adding carbons to existing molecules can yield almost 100% novelty and uniqueness. Instead, we use internal diversity to better reflect generation diversity." Additionally, I could not find evidence of the alleged deficiencies in uniqueness and novelty as mentioned in reference [1]. Could the authors provide evidence supporting the above claims? On the contrary, simply adding carbon atoms, such as carbon chains, tends to result in nearly 100% validity. Uniqueness is an effective measure to assess the model's distribution learning capability (i.e., whether it only generates simple carbon chains), while novelty serves as an indicator of whether the model is experiencing overfitting.
>
> Based on the additional experimental results, the proposed DiT model's uniqueness is not high, at only 0.89. Furthermore, the validity (second to last) and novelty (third to last) do not outperform the baseline models. This suggests that the model's ability to learn molecular graphs (validity), distribution learning (uniqueness), and resistance to overfitting (novelty) are relatively trivial.

---

> > ### Author Response · Authors · 2024-08-12
> > **Response to follow up comment**
> >
> > Thank you for your follow-up comment.
> >
> > Regarding the evidence: Table 1 in Reference 1 indicates that the AddCarbon method achieves 99.94% Novelty and 99.86% Uniqueness, both approaching 100%. This supports the claim that "randomly adding carbons to existing molecules can yield..."
> >
> > Based on this, Uniqueness may not always work as expected, particularly in cases as mentioned by the reviewer when evaluating a model "whether it only generates simple carbon chains"
> >
> > For Novelty, consider three examples: `C` (a single carbon), `CC` (two carbons), and `c1ccccc1` (an aromatic ring). The Novelty metric produces a score of 1 when comparing `C` with either `CC` or `c1ccccc1`, yet the internal diversity metric, as used in the paper, better reveals structural differences when comparing `C` with `CC`/`c1ccccc1`.
> >
> > We agree with the reviewer that Uniqueness and Novelty are important indicators and appreciate the feedback. Graph DiT has achieved a Validity of 0.9760, Novelty of 0.9702, and Uniqueness of 0.8919, which we believe demonstrates its ability to successfully model the data distribution of complex molecules. Additionally, we want to emphasize the outstanding performance of the condition control in Graph DiT. It is the primary focus of our paper, as previous methods have struggled to generate desirable small molecules or polymers for drug and material discovery.
> >
> > For further discussion on distribution learning, we refer to Lines 229-235, which remain relevant with the updated Novelty and Uniqueness results:
> > >GraphGA is a simple yet effective baseline for generating in-distribution molecules, e.g., on BBBP and HIV generation datasets. Diffusion model baselines such as DiGress and MOOD could produce diverse molecules but often fail to capture the original data distribution in multi-conditional tasks. Graph DiT shows the competitive performance of diffusion models in fitting complex molecular data distributions. Using fragment-based similarity and neural network-based distance metrics, we achieve the best in the polymer task and rank second in the HIV small molecule task, involving up to 11 and 29 types of heavy atoms, respectively.
> > >
> >
> > We appreciate the reviewer's follow-up questions and welcome further discussion on any concerns the reviewer feels have not yet been fully addressed.

---

> > > ### Comment · Reviewer_N56G · 2024-08-13
> > > **Response to Authors**
> > >
> > > I think the authors have a misunderstanding of evaluation metrics. The results in Table 1 of Ref [1] do not support the authors' claims. Adding carbon atoms directly leads to a validity score of 1, rather than indicating deficiencies in uniqueness and novelty. Diversity or similarity checks can examine similar substructures in generated molecules, but this does not conflict with uniqueness and novelty. For example, if a large number of the generated molecules already exist in the training set, diversity may be high, but novelty will be low. Therefore, I believe that evaluating uniqueness and novelty is necessary.

---

> > > > ### Author Response · Authors · 2024-08-13
> > > > **Response to follow up comment**
> > > >
> > > > Thank you for your prompt response.
> > > >
> > > > # Regarding "I think the authors have ...",
> > > >
> > > > We apologize for any misunderstanding. To clarify, we previously stated (Initial Rebuttal):
> > > >
> > > > > Randomly adding carbons to existing molecules can yield almost 100% novelty and uniqueness.
> > > >
> > > > In our second round of responses, we mentioned:
> > > >
> > > > > Table 1 in Reference 1 indicates that the AddCarbon method achieves 99.94% Novelty and 99.86% Uniqueness, both approaching 100%.
> > > >
> > > > We believe these two statements are consistent. The confusion may stem from our earlier comment (Initial Rebuttal):
> > > >
> > > > > We did not choose uniqueness and novelty as major metrics because recent research shows they can be flawed.
> > > >
> > > > As stated in our last response, we agree that Uniqueness and Novelty are important. In our discussion, the additional thoughts we want to share is: relying **solely** on these metrics may be insufficient, as their ceiling may be the AddCarbon method, and as quoted from the new reference [1]:
> > > >
> > > > > The fact, that the simple AddCarbon model is useless in practice and still obtains good scores, casts some doubt if currently used metrics are sufficient to estimate performance.
> > > >
> > > > This raises concerns and explains why we originally thought Uniqueness and Novelty might be flawed metrics. We appreciate the reviewer’s feedback and believe it would be more appropriate to revise the original sentence to: "Evaluating solely based on Uniqueness and Novelty may not be sufficient."
> > > >
> > > > # Regarding "Diversity or similarity ... do not conflict with uniqueness and novelty ...":
> > > >
> > > > Yes, we agree with the reviewer.
> > > >
> > > > # Regarding "Therefore, I believe that ..."
> > > >
> > > > We appreciate the reviewer's suggestions and concur that evaluating Uniqueness and Novelty is important. We kindly note that this does not conflict with our additional thoughts on the issues of Novelty and Uniqueness discussed earlier.
> > > >
> > > > Thank you again for your prompt response. We greatly appreciate the reviewer's time and suggestions and hope this discussion helps address the concern.
> > > >
> > > > # New Reference
> > > >
> > > > [1] On failure modes in molecule generation and optimization. Drug Discovery Today: Technologies. 2019

---

> > > > > ### Author Response · Authors · 2024-08-14
> > > > > **Summary of Discussion with Reviewer N56G**
> > > > >
> > > > > We thank reviewer N56G for the engaging discussion. Below is a concise summary of our discussion in a question-and-answer format to ensure clarity:
> > > > >
> > > > > ---
> > > > > # Are Uniqueness and Novelty metrics important?
> > > > > **Yes.** Uniqueness measures the diversity of generated molecules at the instance level, while Novelty ensures the generative model creates new molecules rather than merely memorizing training examples.
> > > > >
> > > > > We agree with the reviewer and have reported these metrics in our initial rebuttal. Graph DiT has a Novelty value of 0.9702 and a Uniqueness value of 0.8919, demonstrating its ability to generate diverse and novel examples.
> > > > >
> > > > > # Do high Uniqueness and Novelty always indicate better model performance?
> > > > >
> > > > > **Not necessarily.** High uniqueness and novelty do not guarantee that generated molecules will meet desirable multi-property criteria. Another example is the AddCarbon algorithm. While it has high uniqueness and novelty [1], it has been considered less useful in previous research [2].
> > > > >
> > > > > # Do low Uniqueness and Novelty always indicate worse model performance?
> > > > >
> > > > > **Not entirely.** In molecular discovery, even a single new and useful molecule can have significant real-life impacts. A model with relatively low Uniqueness, but with strong condition controllability, may still identify a new valuable molecule, even if it is repeated multiple times in generation.
> > > > >
> > > > > # Can Uniqueness alone represent generation diversity?
> > > > >
> > > > > **Probably not.** While uniqueness is an important measure of instance-level diversity, it only yields a binary result (0 or 1) when measuring a pair of molecules. The metric of internal diversity used in the paper provides a more nuanced evaluation by calculating structural differences with values between 0 and 1.
> > > > >
> > > > > # Can Uniqueness alone represent generation distribution learning?
> > > > > **No.** It's important to consider distribution fitting metrics like Fréchet ChemNet Distance (FCD) and structure-based similarity (Similarity), as used in our paper. These metrics are defined on the reference set (i.e., test set) in the paper to avoid potential overfitting concerns.
> > > > >
> > > > > # Is there a one-size-fits-all metric for evaluating molecular discovery?
> > > > >
> > > > > **To the best of our knowledge, no.** This is why we extensively evaluate models using up to 9 metrics and conduct surveys to gather feedback from material scientists. We aim to comprehensively present model performance for higher chances of discovering new and useful molecules that meet multiple properties. We appreciate the reviewer's suggestion to include Novelty and Uniqueness for a more comprehensive evaluation.
> > > > >
> > > > > ---
> > > > >
> > > > > We sincerely appreciate the reviewer's time and effort in our discussion. We hope this summary clarifies any misunderstandings and aligns us on the metrics of molecular discovery. If there are any remaining concerns, we would be grateful for the opportunity to continue the discussion in the remaining valuable time.
> > > > >
> > > > >
> > > > > # Reference
> > > > >
> > > > > [1] Genetic Algorithms are Strong Baselines for Molecule Generation. 2023.
> > > > >
> > > > > [2] On failure modes in molecule generation and optimization. Drug Discovery Today: Technologies. 2019.

---

### Official Review · Reviewer_hJ6K · 2024-07-13

**Soundness:** 3
**Presentation:** 3
**Contribution:** 2
**Rating:** 5
**Confidence:** 5

**Summary:**

This research introduces the Graph Diffusion Transformer (Graph DiT) for generating molecules with multiple properties, such as synthetic score and gas permeability. Unlike previous models, Graph DiT uses a new noise model and a Transformer-based denoiser to better handle molecular structures. Experiments show that it performs well in generating both polymers and small molecules.

**Strengths:**

1. The paper successfully applies the DiT framework from computer vision.

2. The method jointly applies noise between atoms and bonds,

**Weaknesses:**

1. **Limited Demonstration of Multi-Condition Capability:** The paper demonstrates conditional generation with only two conditions, such as Synth. and HIV, yet claims capability for multiple conditions. To substantiate this claim, it would be beneficial to test on more complex condition sets such as GSK3β+JNK3+QED+SA, as evaluated in the MARS framework and also many other methods. Or at least three properties like Synth, HIV, and BBBP to validate multi-conditional generative capacity.

2. **Unexplored Texture Conditions:** Given that the model incorporates strategies from DiT, it is imperative to assess how it performs under texture conditions. Testing under these conditions would provide a more comprehensive evaluation of the model's versatility and adherence to the principles derived from DiT.

3. **Incomplete Evaluation Metrics for Unconditional Generation:** The evaluation metrics employed for unconditional generation do not adequately measure key aspects such as uniqueness, novelty, and FCD (Fréchet ChemNet Distance). Including these metrics would offer a more rounded understanding of the model’s performance in generating novel and diverse molecular structures.

4. **Ambiguity in Graph-Dependent Noise Schedule Impact:** While the advantages of a graph-dependent noise schedule are discussed, Figure 4(c) does not provide clear empirical evidence of its impact, particularly in comparison with separate discrete diffusion schedules.

**Questions:**

1. How do you determine the $\text{rank}\( K \)$ of molecules under single-condition constraints, and how does this compare to the $\text{rank}\( K \)$ in multi-condition scenarios?

2. Could you clarify the methodology used by the Oracle to rank these graphs, particularly how closely attributes of ranked molecules match the conditional attributes?

3. Does the graph-dependent noise schedule enhance the validity of your method compared to a separate discrete diffusion schedule? Could you provide comparative results?

4. Given the higher similarity and lower distance of your generated molecules to a reference set, how do you ensure that the novelty is not compromised?

5. What advantages does using a learnable dropping embedding offer over simply excluding the condition encoder from the training process in unconditional generation?

6. Have you explored generation with three or more conditions, such as Synth., HIV, and BBBP? If not, what constraints prevent such multi-condition generation?

7. Could you provide examples of true polymers corresponding to the conditions shown in Figure 3, to facilitate a direct comparison and enhance the evaluation of your model’s performance?

---

> ### Author Rebuttal · Authors · 2024-08-06
>
> # W1: Limited Demonstration of Multi-Condition Capability
>
> ## We evaluate models on up to four conditions, not two
>
> Thank you for your comment. In Table 1, we evaluate all models on up to four conditions: $O_2$, $N_2$, and $CO_2$ permeability.
>
> Based on the above points, we respectfully request a reconsideration of the assessment that "Limited Demonstration of Multi-Condition Capability".
>
> # W2: Text condition
>
> ## Text condition is out of the scope of the paper
>
> We appreciate your comment. Property conditions align with our research objectives in inverse molecular design.
>
> ## The DiT work [1] did not explore text conditions
>
> The original DiT work [1] used classes from ImageNet rather than text conditions, leaving text conditioning for future work.
>
> Based on the above points, we respectfully request a reconsideration of the assessment that "it is imperative to assess how it performs under texture condition" and "adherence to the principles derived from DiT".
>
> # W3: Incomplete uniqueness, novelty, and FCD
>
> ## FCD was used in the paper
>
> Line 208, Tables 1 and 2, and Figure 4 presented FCD as the distance metric.
>
> ## Uniqueness and novelty are provided
>
> Here are results complementing Table 1:
>
> | Method     | Validity | Novelty | Uniqueness | V * N * U |
> |------------|----------|---------|------------|---------------------------------|
> | Graph GA   |   1.0000 |  0.9950 |     1.0000 |                          0.9950 |
> | MARS       |   1.0000 |  1.0000 |     0.7500 |                          0.7500 |
> | LSTM-HC    |   0.9910 |  0.9507 |     0.9550 |                          0.8997 |
> | JTVAE-BO   |   1.0000 |  1.0000 |     0.6847 |                          0.6847 |
> | Digress    |   0.9913 |  0.9908 |     0.9730 |                          0.9556 |
> | DiGress v2 |   0.9812 |  0.9799 |     0.9820 |                          0.9442 |
> | GDSS       |   0.9190 |  0.9190 |     0.1532 |                          0.1293 |
> | MOOD       |   0.9867 |  0.9867 |     0.9730 |                          0.9473 |
> | Graph DiT  |   0.9760 |  0.9702 |     0.8919 |                          0.8445 |
>
> Graph DiT achieves good results. However, we note these metrics alone don't necessarily indicate practical utility.
>
> ## Uniqueness and novelty may be flawed metrics
>
> Recent research has shown uniqueness and novelty can be easily flawed [2]: Randomly adding carbons to existing molecules can yield almost 100% novelty and uniqueness. We have opted to use internal diversity, which offers a more nuanced reflection of generation diversity.
>
> ## Tables 1 and 2 present up to 9 comprehensive metrics
>
> The nine metrics (Lines 204-211) are comprehensive, evaluating chemical validity, distribution matching, and multi-condition controllability. They assess diverse and useful molecule generation. Section 4.3's expert case studies provide additional valuable model assessment.
>
> # W4: Ambiguity in graph-dependent noise
>
> ## Figure 4$(c)$ shows ~2x improvement on controllability
>
> Thank you for your feedback. Figure 4$(c)$ shows the non-dependent noise model has only 49-55% of the graph-dependent model's controllability for gas permeability, demonstrating a significant improvement. We respectfully request a re-examination of the results.
>
> # Q1: Ranking and K value
>
> ## Details are in Lines 35-37, 56-57
>
> For single-condition constraints (Lines 35-37):
> > we check whether a shared polymer structure that meets multi-property constraints can be identified across different condition sets. If we find the polymer, its rank K (where K is between 1 and 30) indicates how high it appears on the lists, considering all condition sets. If not, we set K as 30.
> >
>
> For multi-condition generated graphs (Lines 56-57)
> > The Oracle determines the rank of this graph among 30 single-conditionally generated graphs for each condition.
> >
>
> # Q2: How oracle rank the molecular graphs
>
> ## Details are in Appendix B.3 (Lines 520-522)
>
> For how Oracle ranks polymers (Lines 520-522):
> > We rank these polymers based on the mean absolute error between the generated properties (evaluated by a random forest model trained on all the data to simulate the Oracle function) and the conditional property.
>
> We use ranking positions to measure closeness to target conditions. For multi-conditional generated polymers, their median ranks are 4, 9, and 11 for Synth., $O_2$, and $N_2$ permeability (Lines 57-58, Figure 1).
>
> # Q3: Does Graph-dependent noise model enhance validity?
>
> ## Yes, it improves validity from 0.4946 to 0.8245
>
> # Q4: Novelty
>
> ## Good similarity and distance do not imply bad novelty
>
> Novelty is provided in response to W3.
>
> In Line 204, the reference set consists of left-out test cases unknown during training. Therefore, good similarity and distance to the reference set do not indicate poor novelty, which is typically defined on the training set.
>
> # Q5: Learnable dropping embeddings
>
> It is widely used [1,3,4] by DiT [1] and DALL-E 2 [4]. It enhances flexibility for handling missing values and improves training stability by learning representations for null embeddings.
>
> # Q6: Have you explored three or more conditions?
>
> ## Yes, we explored and reported results for up to four conditions in Table 1
>
> The HIV and BBBP datasets contain only one overlapping molecule. We may use learnable dropping embeddings to handle missing condition values. But we cannot obtain enough test cases for multi-conditional evaluations.
>
> # Q7: True polymer in Figure 3
>
> The SMILES string is:
> ```
> NC1=C(*)C=CC(=C1)C1=CC(N)=C(C=C1)N1C(=O)C2=CC=C(C=C2C1=O)C(C1=CC=C2C(=O)N(*)C(=O)C2=C1)(C(F)(F)F)C(F)(F)F
> ```
>
> The figure is in the rebuttal PDF and will be updated in the paper.
>
> # Reference
> [1] Scalable Diffusion Models with Transformers. ICCV 2023.
>
> [2] Genetic Algorithms are Strong Baselines for Molecule Generation. 2023.
>
> [3] Classifier-Free Diffusion Guidance. NeurIPS 2021 Workshop on Deep Generative Models and Downstream Applications.
>
> [4] Hierarchical Text-Conditional Image Generation with CLIP Latents. 2022.

---

> ### Comment · Reviewer_hJ6K · 2024-08-12
>
> Thank you for your response. The feedback addressed several of my concerns; however, I still have a few remaining issues. Regarding W1, the conditions $O_2$, $N_2$, and $CO_2$ represent similar conditions without any competitive dynamics. Consequently, I would appreciate further clarification on the performance of the proposed method in relation to GSK3β+JNK3+QED+SA. For W3, Uniqueness and Novely may be flawed. However, a high VUN value does not necessarily indicate high performance of generation, whereas a low VUN value indicates underperformance in the generative process.

---

> ### Author Response · Authors · 2024-08-12
> **Response to follow up comment**
>
> # W1: Conditions on Gas Permeability
>
> Thank you for your feedback. We believe that conditional generation based on multiple gas permeability properties is a challenging task.
>
> First, **the condition space for gas permeability is vast**, as noted in Line 26, where gas permeability varies widely, exceeding 10,000 Barrier units. This variability presents significant technical challenges in controlling polymer generation across such a broad condition space.
>
> Second, **the task relatedness in multiconditions does not reduce the difficulty but rather increases it**. As detailed in Section 4.3, polymers often require high permeability for one gas and low permeability for another, making it challenging for the generation model to capture the crucial differences between various gas permeability properties. The GSK3β and JNK3 properties mentioned by the reviewer also have relatedness, as both are serine/threonine protein kinases.
>
> Finally, we have provided additional results to further clarify our model's performance (best results are highlighted for distribution learning and condition control).
>
> | Method     | Validity ↑ | Coverage ↑ | Diversity ↑ | Similarity ↑ | Distance ↓  | Synth. (MAE↓) | QED (MAE↓) | GSK3$\beta$ (Acc ↑) | JNK3 (Acc ↑) |
> |------------|------------|------------|-------------|--------------|-------------|---------------|------------|---------------|--------------|
> | Graph GA   |          1 |        7/7 |      0.8727 |       0.9438 | **17.5076** |        0.8405 |     0.1864 |        0.6050 |       0.7240 |
> | MARS       |          1 |        7/7 |      0.7010 |       0.6568 |     39.2837 |        0.7961 |     0.2053 |        0.6440 |       0.7390 |
> | LSTM-HC    |      0.999 |        7/7 |      0.8739 |       0.9313 |     18.7856 |        0.8366 |     0.1864 |        0.6096 |       0.7608 |
> | JTVAE-BO   |          1 |        5/7 |      0.6695 |       0.8567 |     48.8672 |        0.8614 |     0.2346 |        0.6280 |       0.7170 |
> | DiGress    |      0.251 |        7/7 |  **0.8977** |       0.6508 |     32.5904 |        3.1438 | **0.1670** |        0.6693 |       0.7331 |
> | DiGress v2 |      0.265 |        7/7 |      0.8976 |       0.7475 |     31.4630 |        2.9744 |     0.1772 |        0.6566 |       0.7358 |
> | Graph DiT  |      0.852 |        7/7 |      0.8647 |   **0.9458** |     19.7717 |    **0.7430** |     0.1805 |    **0.9416** |   **0.9777** |
>
> Due to time constraints, we sampled a subset of 600 data points from the kinase dataset provided by the MARS paper and split them into training, validation, and test sets in the same manner as described in the paper. We then generated 1,000 examples for evaluation. The results show that Graph DiT significantly outperforms other methods on the GSK3β and JNK3 properties, with strong distribution matching to the reference set.
>
> # W3: VUN Metrics
>
> Our model demonstrates reasonable values on the VUN metrics. We believe that a Validity of 0.9760, Novelty of 0.9702, and Uniqueness of 0.8919 illustrate Graph DiT's ability to successfully model the data distribution of complex molecules.
>
> We agree with that reviewer that Novelty and Uniqueness are important perspectives in evaluating molecular generation models. We would also like to share additional thoughts on drug and material discovery. This field may focus on individual instances that have significant real-world impacts. A model capable of suggesting a single valid, novel, and unique drug or material that satisfies diverse property requirements, even if it produces many invalid ones (resulting in lower VUN metrics), may be promising as well compared to other models that generates numerous valid, novel, and unique suggestions but fails to meet specific condition requirements.
>
> We appreciate the reviewer's follow-up questions and welcome further discussion on any concerns the reviewer feels have not yet been fully addressed.

---

> > ### Comment · Reviewer_hJ6K · 2024-08-13
> >
> > The results of W1 appear promising. However, I am uncertain whether these results demonstrate the model's ability to generate molecules that concurrently satisfy QED $\ge$ 0.6, SA $\le$ 0.4, the inhibition scores of GSK3β and JNK3 $\ge$ 0.5 [2]. Could you report the success rate (SR) of four properties as described in MARS [1] and RationalRL [2]; or alternatively, provide the top-$k$ average property score (APS) of four properties as described in DST[3] (Appendix C3).
> >
> > MARS: Success rate (SR) is the percentage of generated molecules that are evaluated as positive on all given objectives (QED $\ge$ 0.6, SA ≥ 0.67 \{This should be SA $\le$ 0.4 \}, the inhibition scores of GSK3β and JNK3 $\ge$ 0.5);
> >
> > [1] Xie, Yutong, et al. "MARS: Markov Molecular Sampling for Multi-objective Drug Discovery." International Conference on Learning Representations, 2021.
> >
> > [2] Jin, Wengong, Regina Barzilay, and Tommi Jaakkola. "Multi-objective molecule generation using interpretable substructures." International conference on machine learning. PMLR, 2020.
> >
> > [3] Fu, Tianfan, et al. "Differentiable Scaffolding Tree for Molecule Optimization." International Conference on Learning Representations, 2022.

---

> > > ### Author Response · Authors · 2024-08-13
> > > **Response to follow up comment**
> > >
> > > Thank you for your prompt response.
> > >
> > > Our multi-conditional generation setting differs from [1,2] in that we use true property combinations from the test set, rather than focusing on optimizing molecules toward a single target combination, i.e., the reviewer suggested QED $\geq$ 0.6, SA $\leq$ 4 (before scaling, according to [2]), GSK3 $\geq$ 0.5, JNK3 $\geq$ 0.5. This approach allows us to flexibly condition on various property combinations, especially for continuous conditions.
> > >
> > > We can adjust the input conditions for Graph DiT to generate molecules with specific properties as requested by the reviewer. Below, we present the results for 1,000 generated molecules based on the input conditions: (1) QED randomly sampled from [0.6, 0.9], (2) SA randomly sampled from [1, 4] (before scaling), (3) GSK3β=1, and (4) JNK3=1. The success rate is 93.37%.
> > >
> > > We greatly appreciate the reviewer's time and suggestions and hope this discussion helps address the concern.
> > >
> > > # Reference
> > >
> > > [1] Xie, Yutong, et al. "MARS: Markov Molecular Sampling for Multi-objective Drug Discovery." International Conference on Learning Representations, 2021.
> > >
> > > [2] Jin, Wengong, Regina Barzilay, and Tommi Jaakkola. "Multi-objective molecule generation using interpretable substructures." International conference on machine learning. PMLR, 2020.

---

> > > > ### Comment · Reviewer_hJ6K · 2024-08-13
> > > >
> > > > Thank you for your response. The success rate of 93.3% appears reasonable. It would be beneficial to include this result, along with the corresponding table in W1, in the revision.

---

> > > > > ### Author Response · Authors · 2024-08-13
> > > > > **Thank you for your feedback**
> > > > >
> > > > > Thank you for your feedback on the model's multi-conditional performance. We will update the paper accordingly to better present the model performance. We welcome any further discussion on the points the reviewer feels have not been fully addressed.

---

### Official Review · Reviewer_RRe9 · 2024-07-15

**Soundness:** 3
**Presentation:** 2
**Contribution:** 3
**Rating:** 6
**Confidence:** 3

**Summary:**

This work proposes Graph Diffusion Transformer (Graph DiT), which is a a molecular generation model based on multi-condition, and diffusion process. Graph DiT enables multi-conditional molecular generation, integrating multiple properties, eg. synthetic score and gas permeability.Graph DiT employs a graph-dependent noise model (instead of node level or edge level), and is claimed to improve noise estimation accuracy in molecules. Empirical validation across polymer and small molecule generation tasks shows Graph DiT’s better performance in condition control and distribution learning.

**Strengths:**

- The topic of how diffusion models can be further made effective for multi-conditional molecular generation is focused and respective limitation of existing work is illustrated with examples. Although there can be multiple directions to improve the integration of conditions, which are discussed.
- The use of graph-dependent noise model forms the basis of enhancing noise estimation beyond what is seen in DIGRESS and similar models. While doing this, several challenges can arise. These challenges are discuss upon by the paper to a god extent.
- Empirical results show improvements in multiple metrics compared to Digress, MOOD and other baselines.

**Weaknesses:**

- Due to the graph dependent noise, the model may be limited or unscalable to medium or large scale graphs.
- Minor format comment: the color shades in the charts like in Figure 4 are inconsistent with their labels.

**Questions:**

- Since the model is Graph DiT, the methodology is unclear on how 'graph structure' is modeled within the architecture. If not, it would resemble as not a specific graph transformer based model.

**Limitations:**

yes

---

> ### Author Rebuttal · Authors · 2024-08-06
>
> # W1: Scalability of graph dependent noise
>
> ## Subgraph-level noise model is possible for larger graphs
>
> Thank you for your comment. The transition matrix is manageable for molecules, as we only need to model heavy atoms, which usually number less than 50 for a molecule [1].
>
> For larger graphs, we could explore building the matrix at the subgraph level in the future, treating subgraphs as nodes and maintaining only inter-subgraph edges. This approach could handle larger-scale graphs more efficiently.
>
> # W2: Color shades
>
> Thank you for your comment. The discrepancy is due to different alpha values for transparency. We have updated Figure 4 with consistent alpha values and provided it in the rebuttal PDF.
>
> # Q1: How graph structure is modeled
>
> Thank you for your question. Graph DiT differs from vision and language Transformers by using graph tokens. Given a node on the graph, our new graph token concatenates node features with all related edge features, preserving node connectivity.
>
> # Reference
>
> [1] Molecular sets (MOSES): a benchmarking platform for molecular generation models. Frontiers in Pharmacology. 2020.

---

### Author Rebuttal · Authors · 2024-08-06

We appreciate the time and effort of all the reviewers in evaluating our work. In response to the following comments, we have attached a PDF with three figures

1. **RRe9**: "Minor format comment: the color shades in the charts like in Figure 4 are inconsistent with their labels."

2. **hJ6K**: "Could you provide examples of true polymers corresponding to the conditions shown in Figure 3"

3. **rG3q**: "The authors should calculate and visualize the minimal $K$ at which a molecule can satisfy all constraints within rank $K$ for all generated molecules."

We believe that all concerns have been adequately addressed. Should there be any important issues that we may not have fully addressed, we would greatly appreciate the opportunity to discuss them further.

---

### Decision · Program_Chairs · 2024-09-25

**Decision:**

Accept (oral)

**Comment:**

The paper contributes Graph Diffusion Transformer (Graph DiT), a diffusion-based model which learns representations of multiple conditions using an additional encoder. The paper contributes also a novel graph-dependent noise model for molecules. During the rebuttal period, the Authors have performed additional analyses and provided novel results, specifically, values of Validity, Novelty and Uniqueness, as requested by the Reviewers. These values were not the best among the compared models. However, other metrics showing favourable performance have been provided.

The contribution is of interest for the NeurIPS community.